# CUMath: A Benchmark and Evaluation Framework for LLMs on Mathematical Reasoning in Undergraduate Computational Math

## Abstract

Large Language Models (LLMs) perform well on popular math benchmarks but still struggle with fundamental undergraduate tasks such as basic integrals. This suggests a diagnostic gap: existing datasets are either trivial, synthetic, or overly advanced, limiting their usefulness for exposing reasoning failures. To address this, we introduce CUMath, a benchmark of 2,100 real problems from undergraduate courses in Calculus, Linear Algebra, Differential Equations, and related fields. Each problem includes step-by-step solutions, enabling evaluation of both final answers and intermediate reasoning. Moreover, current evaluations treat accuracy and reasoning separately, overlooking their joint role in problem-solving. To address this, we propose a multi-layered evaluation framework that combines automatic metrics with an LLM-as-a-grader pipeline, integrating symbolic encoding and external verification. Using this setup, we evaluate 15 LLMs across various prompting strategies. Our results show that even advanced models often misuse symbolic methods and rely on shortcuts, leading to polished but flawed solutions. Our findings reveal the ongoing issue of inconsistent reasoning, highlighting the need for improved benchmarks, evaluation frameworks, and the development of models with enhanced consistency and reasoning capabilities. The code and data will be available upon publication.

## 1 Introduction

Large Language Models (LLMs) have demonstrated remarkable capabilities across diverse domains, including academic question answering and programming (Chen et al., 2021; Hendrycks et al., 2021a). Despite this progress, a persistent gap remains: LLMs continue to struggle with symbolic and multi-step reasoning (Malek et al., 2025), making mathematics one of the most challenging domains in artificial intelligence (Wang et al., 2025; Forootani, 2025). Unlike text-based tasks, mathematics requires not only factual recall but also procedural fluency and logical consistency, elements that remain difficult even for the most advanced systems (Chollet, 2019; Glazer et al., 2024).

Significant progress has been achieved through prompting strategies like Chain-of-Thought (CoT) (Wei et al., 2023) and math-specific pretraining (Peng et al., 2021; Zhou et al., 2023). However, current benchmarks and evaluations are becoming limited. Widely used datasets such as GSM8K (Cobbe et al., 2021) and MATH (Hendrycks et al., 2021b) show near-ceiling performance, while advanced benchmarks like HARDMath (Fan et al., 2024) result in uniformly low scores that make it hard to see where the reasoning actually breaks down. Recent undergraduate-level datasets, such as UGMath (Xu et al., 2025), have attempted to address this gap, but still cover many elementary problems and typically lack step-by-step annotations essential for analyzing reasoning.

In contrast, progress in evaluation frameworks has been far more limited. Two primary approaches have emerged, each with its own limitations. Outcome-centric metrics, such as Exact Match and F1 (Cobbe et al., 2021; Hendrycks et al., 2021b), prioritize final-answer accuracy but overlook the reasoning process. Reasoning-aware metrics, including ROSCOE (Golovneva et al., 2023), ReasonEval (Xia et al., 2024), and LLM-as-a-Judge methods (Gu et al., 2025), assess intermediate steps but often overlook overall correctness. These perspectives are rarely integrated, leaving evaluations

unable to distinguish between correct answers derived from flawed reasoning and valid reasoning that breaks down only at the final step.

To address these issues, we make three main contributions:

1. **A new benchmark for undergraduate-level mathematical reasoning.** We introduce CUMath, a dataset of 2,100 problems evenly distributed across seven core subjects, each with detailed step-by-step solutions for reasoning-focused evaluation. This balanced coverage ensures that no single subject dominates the dataset.

2. **A multi-layered evaluation framework.** To comprehensively assess LLMs, we propose a framework that integrates automatic metrics (Exact Match, F1, Stepwise Reasoning Score, Validity–Redundancy Score) with LLM-as-a-grader feedback. Our LLM-as-a-grader pipeline combines MathBERT for symbolic encoding, an LLM for step-level reasoning assessment, and Wolfram Alpha for answer verification. This design captures both outcome correctness and reasoning quality, two complementary aspects of mathematical problem solving.

3. **An empirical analysis of LLM reasoning gaps.** Using CUMath and our framework, we show that state-of-the-art LLMs continue to exhibit systematic errors in symbolic manipulation and procedural reasoning, even when producing correct final answers. These findings underscore the importance of evaluating reasoning validity in conjunction with correctness.

Together, CUMath and our evaluation framework establish a principled methodology for benchmarking mathematical reasoning in LLMs, balancing correctness with reasoning quality.

## 2 RELATED WORK

Table 1: Comparison of math datasets by level (E: Elementary to Middle School, H: High School, O: Olympiad, U: Undergraduate), computational undergraduate coverage, number of task types, subjects, test size, free response (FR) answer proportion, and inclusion of step-by-step solutions

| Dataset | Levels | %CU | #Types | #Subj. | #Test | %FR | Step-by-step |
|---|---|---|---|---|---|---|---|
| GSM8k (Cobbe et al., 2021) | E | 0 | 1 | – | 1k | 0 | No |
| MATH (Hendrycks et al., 2021b) | H,O | 0 | 3 | 7 | 5k | 100 | Yes |
| MiniF2F (Zheng et al., 2022) | E,H,O | 0 | 3 | – | 244 | 100 | Yes |
| MathVerse (Zhang et al., 2024) | H | 0 | 3 | – | 4.7k | 45 | No |
| MathVista (Lu et al., 2024) | E,H,O | 0 | 3 | – | 5k | 46 | No |
| MATH-V (Lu et al., 2024) | E,H,O | 0 | 3 | – | 3k | 50 | No |
| MMLU$_{\text{Math}}$ (Wang et al., 2024) | E,H,U | 0 | 1 | 3 | 1.3k | 0 | No |
| MathOdyssey (Fang et al., 2024) | H,O,U | ∼10 | 1 | – | 387 | 100 | No |
| MMMUMath (Yue et al., 2024) | E,H,U | 0 | 1 | – | 505 | 0 | No |
| We-Math (Qiao et al., 2024) | H,U | ∼20 | 3 | – | 1.7k | 100 | No |
| OCWCourses (Lewkowycz et al., 2022) | U | ∼18 | 1 | – | 272 | 100 | No |
| ProofNet (Azerbayev et al., 2023) | U | 0 | 1 | – | 371 | 100 | No |
| UGMathBench (Xu et al., 2025) | U | ∼50 | 10 | 16 | 5.5k | 0 | No |
| **CUMath** | **U** | **100** | **3** | **7** | **2.1k** | **∼75** | **Yes** |

**Mathematical Benchmark.** Mathematical reasoning is a key test of LLMs' generalization and problem-solving ability, driving the creation of numerous benchmarks. Early datasets, such as GSM8K (Cobbe et al., 2021) and MATH (Hendrycks et al., 2021b), remain widely used, but primarily cover grade-school word problems and competition-style questions. With models now surpassing 97% on GSM8K and 94% on MATH (Zhou et al., 2023; OpenAI, 2024), these benchmarks are reaching a capability threshold and fail to capture deeper reasoning skills.

Recent datasets, such as GHOST (Frieder et al., 2023), HARDMath (Fan et al., 2024), and ARB (Sawada et al., 2023), introduce more advanced problems, but often lead to uniformly low scores. While effective at exposing limitations, this difficulty gap reduces diagnostic value, as consistent

failure hides specific reasoning weaknesses. Therefore, there is a need for benchmarks that are challenging yet feasible, aligned with current LLM capabilities, while also revealing reasoning gaps.

Undergraduate-level benchmarks, such as UGMath (Xu et al., 2025) and MathOdyssey (Fang et al., 2024), aim to bridge this gap by covering a broad spectrum of topics. However, these datasets include many elementary problems (arithmetic and basic algebra) that are already well-covered in MATH and GSM8K and can be easily handled by current models. Moreover, they typically emphasize final answers over reasoning and lack detailed step-by-step annotations. Therefore, it reduces their usefulness for evaluating advanced reasoning. Meanwhile, computational mathematics—requiring symbolic manipulation and multi-step procedures—remains underrepresented (see Table 1), despite being a central challenge for LLMs (Cao et al., 2025; Mirzadeh et al., 2024).

**Evaluation Frameworks and Reasoning Metric.** Early evaluations of LLMs in mathematics have primarily relied on metrics such as Exact Match and F1 score (Hendrycks et al., 2021b; Cobbe et al., 2021), which assess only final-answer correctness. However, as LLMs now achieve near-human performance on GSM8K and MATH (Zhou et al., 2023; OpenAI, 2024), these outcome-focused metrics are reaching a capability threshold and fail to capture the quality of reasoning.

To overcome these limitations, researchers have begun to develop reasoning-aware evaluation frameworks. For example, the ROSCOE suite (Golovneva et al., 2023) measures reasoning chains along dimensions such as faithfulness, coherence, and informativeness, producing scores that align more closely with human judgment. Building on this, ReasonEval (Xia et al., 2024) assesses validity and redundancy at the step level, enabling more fine-grained analysis of reasoning quality. Other efforts adopt the LLM-as-a-Judge paradigm (Gu et al., 2025), where stronger models grade reasoning traces and achieve strong agreement with human experts. Broader frameworks, including MMLU-Pro+ (Taghanaki et al., 2024), extend evaluation to multi-dimensional reasoning, while UGMathBench (Xu et al., 2025) introduces multi-version testing to assess robustness.

**Improving Mathematical Reasoning in LLMs.** Beyond benchmarking and evaluation, a parallel line of work focuses on improving the reasoning capabilities of LLMs themselves. One direction explores prompting strategies such as Chain-of-Thought (CoT) (Wei et al., 2023), Tree-of-Thoughts (ToT) (Yao et al., 2023), and Self-Consistency (SC) (Wang et al., 2023), which encourage structured reasoning traces. Another direction involves model-level adaptation, including fine-tuning on curated datasets (Zhou et al., 2023) and continued pretraining on math-specific corpora (Peng et al., 2021), leading to specialized math models. Despite this progress, LLMs still frequently hallucinate intermediate steps, misuse operations, or fail on symbolic manipulation (Cao et al., 2025; Malek et al., 2025). Crucially, these errors can occur even when the final answer is correct, highlighting the persistent gap between surface accuracy and genuine reasoning ability. This mismatch underscores the need for evaluation methods that extend beyond outcome correctness and directly assess the quality of reasoning processes.

To address these gaps, we introduce CUMath, a balanced benchmark for undergraduate mathematical reasoning, together with a multi-layered evaluation framework, and use them to reveal systematic reasoning gaps in state-of-the-art LLMs.

## 3 CUMATH DATASET

We present the CUMath dataset, a benchmark for assessing mathematical reasoning in undergraduate mathematics. Unlike existing datasets that focus on artificial or competition-style problems, CUMath is derived from actual instructional materials, reflecting the reasoning challenges that undergraduate students encounter.

The dataset consists of 2,100 problems evenly distributed across seven core areas of undergraduate computational mathematics: Calculus, Differential Equations, Discrete Mathematics, Linear Algebra, Multivariable Calculus, Precalculus, and Trigonometry, with each area containing exactly 300 problems. Unlike previous datasets that often overrepresented certain domains, such as calculus or elementary algebra, this balanced distribution prevents topic bias. This enables a fair comparison of model performance across different topics and supports a more comprehensive assessment of mathematical reasoning. We categorize the problems into three answer formats: Free Response (FR), Short Answer (SA), and True/False (TF). For each problem, CUMath provides detailed, step-by-step solutions, enabling a comprehensive evaluation of understanding that extends beyond simply

checking for the final answer's accuracy. A breakdown of problem distribution by sub-topics is provided in Appendix B. Our CUMath creation process consists of three phases: data collection, data cleaning and formatting, and data labeling.

**Data Collection.** CUMath problems are drawn from two primary sources: (i) [anonymized] university quizzes, exams, and problem sets, and (ii) open-access textbooks that are widely recommended by American Institute of Mathematics and protected by Creative Commons licenses. Closed materials have been included through instructors' agreements (see Appendix A for details). While we can't guarantee these materials were excluded from LLM training data, licensing restrictions and the private nature of quizzes and exams reduce this likelihood. Math educators reviewed all problems for clarity and correctness, preserving the original wording and notation. The datasets were originally in LaTeX or PDF format and are released for non-commercial use only.

**Data Cleaning and Formatting.** Each problem was standardized into a structured JSON format to enable consistent access and downstream use. During this phase, text was cleaned to correct typographical errors and remove formatting artifacts. Mathematical expressions were encoded in LaTeX to ensure proper rendering and compatibility with language model input formats. We performed deduplication to eliminate redundant problems, ensuring each issue was self-contained and isolated from the surrounding content.

**Data Labeling.** We annotated each problem with metadata to support fine-grained analysis and structured evaluation. The core fields include a unique identifier, topic and subtopic labels, question text, source attribution, and expected response format. To support both coarse- and fine-grained evaluation, entries include a final answer and a step-by-step solution. Each problem is categorized into one of three response types: FR, SA, and TF, reflecting the typical assessment styles used in mathematics courses. Examples of annotated problems are provided in Appendix C.

## 4 EVALUATION METRICS FRAMEWORK

We assess model performance by integrating 4 different automatic metrics (Accuracy, Semantic F1, Stepwise Reasoning Score, and Validity–Redundancy Score) with LLM-as-a-grader feedback for a comprehensive assessment of final-answer correctness and step-by-step reasoning quality.

### 4.1 AUTOMATIC METRICS

**Evaluation Formulation.** Let $\mathcal{D} = (q_i, a_i)$ be the CUMath dataset, where $q_i$ denotes the problem statement and $a_i$ denotes ground-truth answers, and $S_i = \{s_i^1, \ldots, s_i^{n_i}\}$ the corresponding reference reasoning steps. Consider a LLM represented as $M$, denote its predicted final answer $\hat{a}_i = M(q_i)$ and reasoning steps $\hat{S}_i = \{\hat{s}_i^1, \ldots, \hat{s}_i^{\hat{n}_i}\}$. We additionally denote by $e_i = (e_i^1, e_i^2, \ldots, e_i^{n_i}) = (\hat{s}_i^1, \hat{s}_i^2, \ldots, \hat{s}_i^{\hat{n}_i})$ the same reasoning steps viewed as an ordered sequence. Based on these notations, we define metrics that evaluate both reasoning steps and the correctness of the final answer.

**Accuracy.** To account for algebraic equivalence, both $a_i$ and $\hat{a}_i$ are parsed into symbolic form using SymPy. Denote $\phi(\cdot)$ is the parsing function and $\mathbb{I}[\cdot]$ the indicator. If parsing fails, string matching is used. Correctness is then

$$\text{Accuracy}(M) = \frac{1}{|\mathcal{D}|} \sum_{i=1}^{|\mathcal{D}|} \mathbb{I}\big[\phi(\hat{a}_i) \equiv \phi(a_i)\big],$$

**Semantic F1.** To measure alignment between generated and human reference steps, we compute a semantic F1-Score. Let $\mathcal{X}$ be the set of all possible reasoning steps, and let $f : \mathcal{X} \to \mathbb{R}^d$ be a pretrained encoder. For each problem $i$, the reference steps $s_i^j, \hat{s}_i^k \in \mathcal{X}$ for $j = \overline{1, n_i}$, $k = \overline{1, \hat{n}_i}$. Each step $s_i^j$ and $\hat{s}_i^k$ is mapped via $f$ to embeddings $f(s_i^j), f(\hat{s}_i^k) \in \mathbb{R}^d$. We then compute pairwise similarities between $s_i^j \in S_i$ and $\hat{s}_i^k \in \hat{S}_i$ using cosine similarity:

$$C_{jk} = \cos\big(f(s_i^j), f(\hat{s}_i^k)\big) = \frac{f(s_i^j)^\top f(\hat{s}_i^k)}{\|f(s_i^j)\| \, \|f(\hat{s}_i^k)\|}.$$

A greedy one-to-one matching $\mathcal{M}_i$ is constructed by sorting pairs $(j, k)$ in descending $C_{jk}$ and selecting them if $C_{jk} \geq \tau$ (with $\tau = 0.7$) and neither step has already been matched. Let $M_i = |\mathcal{M}_i|$

denote the number of matched pairs. We compute the dataset-level precision, recall, and F1 as:

$$\text{Precision}(M) = \frac{\sum_{i=1}^{|\mathcal{D}|} M_i}{\sum_{i=1}^{|\mathcal{D}|} |\hat{S}_i|}, \ \text{Recall}(M) = \frac{\sum_{i=1}^{|\mathcal{D}|} M_i}{\sum_{i=1}^{|\mathcal{D}|} |S_i|}, \ \text{F1}(M) = \frac{2 \cdot \text{Precision}(M) \cdot \text{Recall}(M)}{\text{Precision}(M) + \text{Recall}(M)}.$$

**Stepwise Reasoning Score (SRS).** Following the ROSCOE framework (Golovneva et al., 2023), we evaluate reasoning quality using a subset of fine-grained metrics. For each solution, we compute six metrics: Faithfulness, Informativeness (Step), Informativeness (Chain), Coherence (Step vs. Step), Discourse Representation, and Repetition (Step). We denote the score assigned by the $k$-th metric as $m_k(e_i)$, where $k$ ranges from 1 to 6. All metrics are normalized to $[0, 1]$, with higher values consistently indicating better quality (see Appendix D.4 for details). The per-solution score $(\text{SRS}(e_i))$ and the dataset-level score $(\text{SRS}(M))$ will be computed as follows

$$\text{SRS}(e_i) = \frac{1}{6} \sum_{k=1}^{6} m_k(e_i), \quad \text{SRS}(M) = \frac{1}{|\mathcal{D}|} \sum_{i=1}^{|\mathcal{D}|} \text{SRS}(e_i)$$

**Validity and Redundancy (VR).** We adapt *ReasonEval* (Xia et al., 2024), which evaluates reasoning based on per-step validity and redundancy. Each step $\hat{s}_i^j$ is compared with the problem $q_i$ using an NLI model that outputs probabilities for *entailment*, *neutral*, and *contradiction*. From these, $S_j^{\text{validity}} = p_j^{\text{entail}} + p_j^{\text{neutral}}$, and $S_j^{\text{redundancy}} = p_j^{\text{neutral}}$. The per-solution score $(\text{VR-Score}(e_i))$ and the dataset-level score $(\text{VR-Score}(M))$ will be computed as follows

$$\text{VR-Score}(e_i) = \min_j S_j^{\text{validity}} - \max_j S_j^{\text{redundancy}}, \quad \text{VR-Score}(M) = \frac{1}{|\mathcal{D}|} \sum_{i=1}^{|\mathcal{D}|} \text{VR-Score}(e_i).$$

### 4.2 LLM AS A GRADER

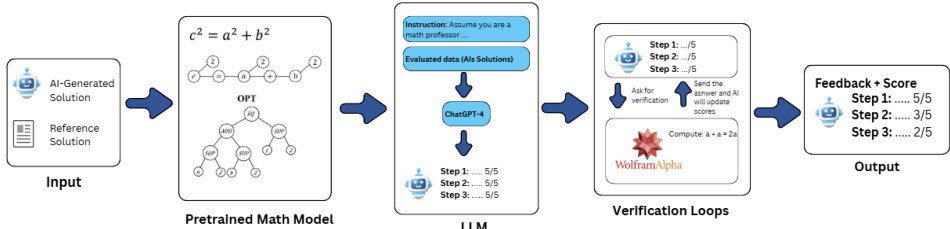

Figure 1: Overview of our grading pipeline. MathBERT encodes expressions, LLM gives step-level feedback, and an external Computer Algebra System (CAS) verifies correctness.

We design an automatic grading pipeline that assesses both the correctness of final answers and the quality of the written solution (Figure 1). This matters because two solutions with the same final answer can come from very different reasoning. Our pipeline takes as input both AI-generated solutions and reference solutions, so each step can be compared against a trusted path.

**Step 1 (Input).** The pipeline begins with both AI-generated and reference solutions, which together provide a basis for comparison against a trusted path.

**Step 2 (Math Segmentation).** To process a student or AI-generated solution, we first perform a step segmentation procedure. Since mathematical solutions are written in free-form text, the pipeline needs a consistent way to isolate units of reasoning. If the solution is explicitly structured with steps labeled such as "step $k$", we use those as natural boundaries. In cases without explicit markers, we default to line-based segmentation, where each line is treated as a candidate reasoning step. Each extracted step is then encoded using MathBERT (Peng et al., 2021), which preserves the structure of equations, improving the accuracy in comparing generated and reference steps.

**Step 3 (LLM Feedback).** The encoded steps, along with a textual representation of their embeddings, are provided to an LLM prompted to act as a mathematical instructor (see Appendix D.3).

The LLM uses this information to deliver step-level feedback by identifying errors, reasoning gaps, and partial correctness. In addition to qualitative comments, the LLM assigns a preliminary score on a 0–5 scale reflecting the validity and clarity of each step.

**Step 4 (Verification Loops).** To improve reliability, we integrate verification loops with external CAS (i.e, Wolfram Alpha). Whenever the pipeline detects that a reasoning step contains a mathematical expression, that expression is extracted, normalized, and sent as a query to the CAS. The CAS then returns the mathematically validated result, such as the simplified form of an equation, the solved solution set, or confirmation of equivalence between two expressions. The pipeline compares the LLM's judgment of the step with the CAS's authoritative output. If the CAS verifies the equivalence, the LLM's proposed assessment is maintained. If a discrepancy arises, for example, when the LLM accepts an invalid manipulation or fails to recognize an equivalence, the CAS result takes priority, and the LLM is prompted to revise its assessment based on the verified computation. This proposer–verifier loop reduces hallucinations, arithmetic mistakes, and symbolic misinterpretations, while ensuring that the grader remains consistent with formal mathematics.

**Step 5 (Output).** The final output is step-level feedback and a numerical score, combining the LLM's reasoning-based assessment with CAS verification to ensure mathematical validity.

## 4.3 Human Validation

To assess the reliability of the LLM-as-a-grader pipeline, we conducted a human evaluation on a subset of the benchmark. We recruited three independent annotators with sufficient mathematical background to evaluate multi-step reasoning at the undergraduate level. All annotators were instructed to use the same scoring rubric as our automatic grading pipeline.

From each of the seven topics, we randomly sampled five problems without replacement to prevent bias and preserve balanced topic coverage. In total, the evaluation set consisted of 35 problems. For each sampled problem, we collected solutions from three representative model families: closed-source, open-source, and math-specialized, each achieving the highest accuracy for that topic. All solutions were generated under CoT prompting. We chose CoT because it achieved the highest performance across Accuracy, F1, SRS, and LLM-based evaluation metrics (as shown later in our experimental evaluation). This setup allowed us to evaluate the grading pipeline using the strongest model outputs observed in our main experiments.

In total, the annotators evaluated 105 solutions. We first assessed human scoring consistency using Krippendorff's $\alpha$ to measure the agreement across all human graders and quadratic-weighted Cohen's $\kappa$ to compute the pairwise agreement between two graders to establish a baseline for different interpretations with the LLM-as-a-grader. We then treated the LLM-as-a-grader as an additional grader and computed the same metrics to measure how closely the LLM aligns with human evaluations.

## 5 Experiments

### 5.1 Experimental Setup

**Evaluated LLMs.**[1] Our evaluation covers 3 categories of LLMs to provide a comprehensive analysis of mathematical reasoning. Closed-source models demonstrate proprietary advancements, open-source models emphasize transparency and community collaboration, and math-specialized models are optimized for symbolic reasoning in targeted assessments. The evaluated LLMs are listed below:

- **Closed-source models:** GPT-4.1, GPT-3.5-turbo-0125, OpenAI o3, Claude Sonnet 3.7.
- **Open-source models:** DeepSeek-R1-Distill-Qwen-32B, Gemma 2 9B IT, LLaMA 3 8B/70B Instruct, LLaMA 4 Scout 17B Instruct, Qwen2.5 7B Instruct, Mistral 7B Instruct v0.3.
- **Math-specialized models:** Qwen2.5-Math-7B Instruct, Qwen2.5-Math-1.5B Instruct, Llemma-7B, LLaMA-3.2-1B Instruct (ft).

Detailed specifications of these models are provided in Appendix D.1.

---

[1]All experiments were conducted within a fixed window (May–August 2025). Throughout the paper, "frontier models" refers to the state-of-the-art models available at the time we conducted our experience.

**Prompting Styles.** We evaluate four prompting techniques commonly used to enhance reasoning in LLMs: Zero-shot, Chain-of-Thought (CoT), Self-Consistency (SC), and Tree-of-Thoughts (ToT). The full set of prompt templates used in our experiments is provided in Appendix D.2

**Evaluation settings.** All models are evaluated using the four prompting techniques described above.

To ensure consistency and reproducibility, we standardize decoding parameters across all models. Specifically, both Zero-Shot and CoT employ greedy decoding with a temperature set to 0, meaning the model deterministically selects the most probable next token at each step. For SC, we sample 5 reasoning chains at temperature 0.9 and select the final answer by majority vote, following Wang et al. (2023). For ToT, we generate 3 distinct reasoning paths at temperature 0.7, following Yao et al. (2023), to encourage exploratory reasoning.

All outputs are constrained to a maximum length of 2,048 tokens. This limit is sufficient to capture the complete reasoning process and final answers for all problems in our dataset, while fitting within the context window of all evaluated models. This ensures consistency across models with different maximum token capacities. We evaluate model performance using automatic metrics (Accuracy, Semantic F1, SRS, and VR) and an LLM-based grading pipeline. To ensure reliability, we additionally conducted a human evaluation of the grading process, which is described in detail in Section 4.

## 5.2 MAIN RESULTS

Final-answer accuracy on CUMath remains significantly lower than benchmarks for grade-school or competition-style math. Even the best LLMs achieve only about 25% accuracy. A closer look by topic shows a clear trend: the harder the topic, the more the models struggle (detailed topic results are in Appendix E). Even the best performance reached only around 10% in sophomore/junior-level courses such as differential equations, multivariable calculus, and linear algebra. By comparison, freshman-level calculus and discrete mathematics reached 25–30% and 15–20%, while introductory topics such as trigonometry and pre-calculus can achieve up to 36% and 42%.

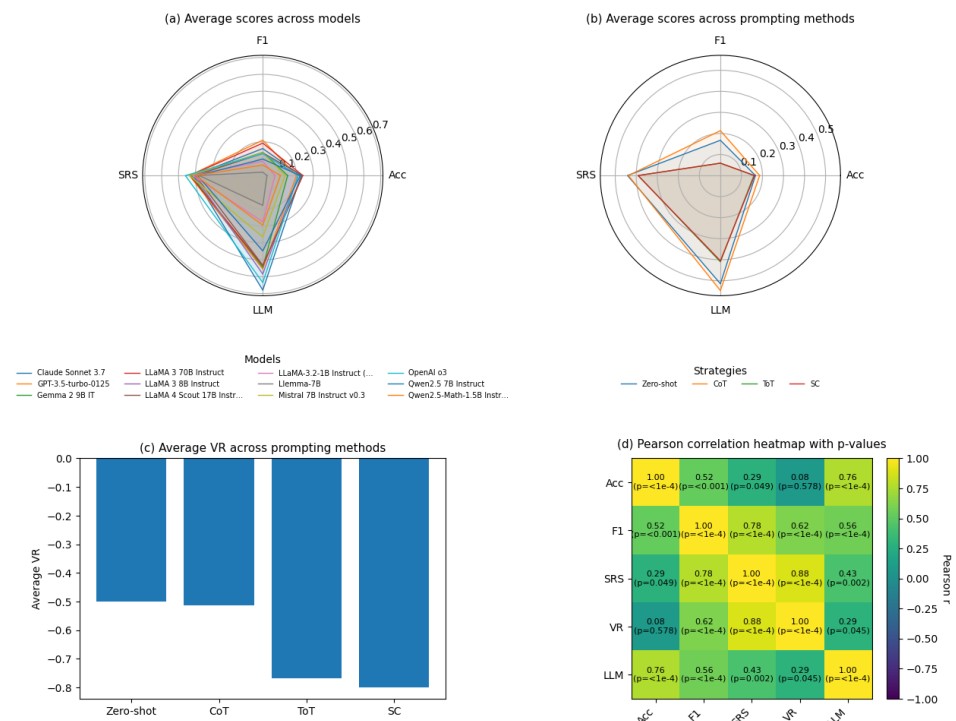

Figure 2: **Multi-metric summary of CUMath results.** (a) Average scores across models. (b) Average scores across prompting methods. (c) Average VR across prompting methods. (d) Pearson correlation heatmap with $p-$values.

Table 2: **Main Results on CUMath.** Evaluation of LLMs across four prompting strategies and five metrics: Accuracy (Acc), Semantic F1, SRS, VR, and LLM-based evaluation (LLM, normalized to [0,1]). The highest value in each column is highlighted in **bold and underlined**

| Model | Zero-shot | | | | | CoT | | | | | ToT | | | | | SC | | | | |
|---|---|---|---|---|---|---|---|---|---|---|---|---|---|---|---|---|---|---|---|---|
| | Acc | F1 | SRS | VR | LLM | Acc | F1 | SRS | VR | LLM | Acc | F1 | SRS | VR | LLM | Acc | F1 | SRS | VR | LLM |
| *Closed-source Models* | | | | | | | | | | | | | | | | | | | | |
| GPT-4.1 | **0.23** | 0.11 | 0.39 | -0.53 | 0.53 | 0.24 | 0.12 | 0.39 | -0.53 | 0.56 | 0.21 | 0.03 | 0.36 | -0.75 | 0.27 | **0.24** | 0.03 | 0.37 | -0.72 | 0.25 |
| GPT-3.5-turbo-0125 | 0.21 | **0.29** | 0.46 | -0.51 | 0.52 | 0.21 | **0.38** | **0.48** | -0.39 | 0.55 | 0.18 | **0.09** | 0.40 | -0.78 | 0.57 | 0.20 | 0.08 | 0.40 | -0.83 | 0.58 |
| OpenAI o3 | 0.21 | 0.22 | **0.51** | **-0.23** | 0.67 | 0.22 | 0.18 | **0.48** | **-0.30** | 0.69 | 0.19 | 0.05 | **0.43** | **-0.44** | **0.60** | 0.22 | 0.07 | **0.42** | **-0.47** | 0.58 |
| Claude Sonnet 3.7 | **0.23** | 0.20 | 0.42 | -0.72 | **0.73** | 0.24 | 0.20 | 0.42 | -0.65 | **0.73** | 0.21 | **0.09** | 0.40 | -0.78 | 0.59 | 0.23 | **0.15** | 0.41 | -0.74 | **0.67** |
| *Open-source Models* | | | | | | | | | | | | | | | | | | | | |
| DeepSeek-R1-Distill-Qwen-32B | 0.22 | 0.10 | 0.40 | -0.68 | 0.63 | **0.25** | 0.10 | 0.39 | -0.69 | 0.55 | 0.21 | 0.04 | 0.38 | -0.88 | 0.31 | 0.20 | 0.02 | 0.38 | -0.89 | 0.21 |
| Gemma 2 9B IT | 0.19 | 0.16 | 0.44 | -0.44 | 0.61 | 0.21 | 0.27 | 0.47 | -0.43 | 0.63 | 0.13 | 0.07 | 0.39 | -0.72 | 0.54 | 0.06 | 0.05 | 0.40 | -0.71 | 0.40 |
| LLaMA 3 8B Instruct | 0.21 | 0.18 | 0.42 | -0.63 | 0.62 | 0.23 | 0.24 | 0.43 | -0.62 | 0.64 | 0.20 | 0.06 | 0.39 | -0.85 | 0.52 | 0.20 | 0.04 | 0.39 | -0.89 | 0.56 |
| LLaMA 3 70B Instruct | **0.23** | 0.26 | 0.49 | -0.34 | 0.41 | **0.25** | 0.35 | 0.46 | -0.45 | 0.61 | **0.23** | 0.08 | 0.39 | -0.82 | 0.57 | **0.24** | 0.08 | 0.39 | -0.83 | 0.55 |
| LLaMA 4 Scout 17B Instruct | **0.23** | 0.18 | 0.41 | -0.70 | 0.67 | **0.25** | 0.23 | 0.41 | -0.66 | 0.65 | **0.23** | 0.08 | 0.39 | -0.83 | 0.44 | 0.23 | 0.05 | 0.39 | -0.85 | 0.37 |
| Qwen2.5 7B Instruct | 0.21 | 0.14 | 0.40 | -0.62 | 0.55 | 0.24 | 0.17 | 0.41 | -0.60 | 0.56 | 0.22 | 0.04 | 0.36 | -0.81 | 0.30 | 0.22 | 0.04 | 0.38 | -0.83 | 0.38 |
| Mistral 7B Instruct v0.3 | 0.12 | 0.15 | 0.48 | -0.32 | 0.29 | 0.17 | 0.27 | 0.46 | -0.43 | 0.40 | 0.10 | 0.06 | 0.40 | -0.81 | 0.37 | 0.14 | 0.06 | 0.40 | -0.86 | 0.40 |
| *Math-specialized Models* | | | | | | | | | | | | | | | | | | | | |
| Qwen2.5-Math-7B Instruct | 0.14 | 0.12 | 0.39 | -0.63 | 0.47 | 0.14 | 0.08 | 0.40 | -0.56 | 0.29 | 0.14 | 0.04 | 0.36 | -0.78 | 0.32 | 0.15 | 0.03 | 0.35 | -0.83 | 0.23 |
| Llemma-7B | 0.03 | 0.03 | 0.41 | -0.48 | 0.23 | 0.03 | 0.03 | 0.40 | -0.48 | 0.28 | 0.02 | 0.01 | 0.36 | -0.82 | 0.09 | 0.02 | 0.01 | 0.34 | -0.94 | 0.11 |
| Qwen2.5-Math-1.5B Instruct | 0.06 | 0.08 | 0.43 | -0.48 | 0.39 | 0.12 | 0.10 | 0.40 | -0.59 | 0.49 | 0.12 | 0.04 | 0.38 | -0.75 | 0.17 | 0.12 | 0.03 | 0.38 | -0.80 | 0.13 |
| LLaMA-3.2-1B Instruct (ft) | 0.07 | 0.12 | 0.43 | -0.54 | 0.48 | 0.07 | 0.14 | 0.43 | -0.56 | 0.33 | 0.08 | 0.04 | 0.40 | -0.79 | 0.16 | 0.07 | 0.04 | 0.40 | -0.84 | 0.13 |

By default, accuracy is computed via symbolic equivalence, such as $x^2$ is the same as $a^2$, rather than raw string matching, so it ignores trivial notational differences. However, only short-form responses that require a fill-in answer, a single number, or a true/false value are evaluated with string matching. In these cases, symbolic checking is unnecessary, and string matching ensures that simple but valid responses are not penalized. Nonetheless, this method can still understate model performance in cases where SymPy fails to parse the output correctly, implicit domain conditions, or alternative valid representations are not captured by simplification. As shown in Figures 2 (a) and (b) and detailed in Table 2, these low accuracy levels are consistent across families and strategies. Critically, higher accuracy does not reliably translate into stronger reasoning: many correct answers were produced through brittle, incoherent, or redundant derivations, as reflected in low VR scores and only moderate SRS.

Different prompting strategies also affected model performance, consistent with prior findings (Zhuo et al., 2024). Across strategies, CoT achieved the best results in Accuracy, F1, and SRS compared to Zero-shot, ToT, and SC, and it also yielded the highest LLM-based evaluation scores for most models. For VR, CoT generally maintained performance comparable to Zero-shot across closed-source, open-source, and math-specialized models. By contrast, ToT and SC offered only marginal or inconsistent gains. In many cases, these strategies increased redundancy without improving accuracy or reasoning coherence, leaving CoT as the most effective prompting method overall.

These patterns are further illustrated in Figure 2 (d). The correlation heatmap underscores a key limitation of using accuracy alone to assess mathematical capability: it shows only a weak correlation with SRS ($r \approx 0.29$, $p = 0.049$) and essentially no correlation with VR ($r \approx 0.08$, $p = 0.578$). This suggests that correct answers can occur without coherent derivations. In contrast, F1, SRS, and VR are strongly correlated, suggesting that they capture aligned but complementary aspects of reasoning quality, including stepwise alignment with references, logical progression, and conciseness. LLM-as-a-grader scores correlate moderately with both outcome-oriented and reasoning-oriented metrics,

indicating that they integrate aspects of both and better approximate comprehensive solution quality. Overall, these results underscore that accuracy alone can misrepresent model competence, highlighting the need for multidimensional evaluation frameworks.

### 5.3 Human–human and human–LLM reliability

To evaluate the reliability of our human evaluation setup and the alignment of the LLM-as-a-grader with human graders, we report inter-rater agreement across human–human and human–LLM comparisons (Figure 3). Detailed results are provided in Appendix F.

**Human–Human Agreement.** Across all solutions, the three human annotators achieve an overall Krippendorff's $\alpha$ of $0.829$, indicating strong reliability. Topic-level agreement is consistently high, with $\alpha$ values ranging from $0.60$ to $0.93$ across domains. Pairwise Cohen's $\kappa$ follows the same pattern, ranging between $0.42$–$0.93$.

**Human–LLM Agreement.** We evaluate the alignment between human annotators and the LLM using the same reliability metrics. The overall Krippendorff's $\alpha$ remains nearly unchanged at $0.832$, indicating that the LLM preserves dataset-level reliability and aligns well with human judgments. Topic-level agreement ranges from $0.62$ to $0.88$, with $\kappa$ values comparable to human–human variability.

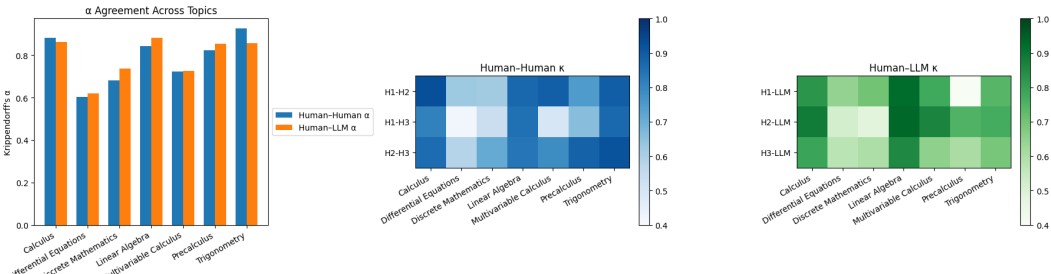

Figure 3: Inter-rater agreement across topics. Left: Krippendorff's $\alpha$ for human–human and human–LLM comparisons. Middle: human–human quadratic Cohen's $\kappa$. Right: human–LLM $\kappa$.

## 6 Where Do Frontier LLMs Still Fail?

Despite recent advances, our analysis shows that frontier LLMs continue to make basic yet systematic errors on undergraduate-level mathematics. These errors are not isolated but show recurring patterns across models and prompting strategies, revealing that core reasoning gaps remain unresolved. Across a wide range of problems, we consistently observe two characteristic failure modes: (1) wrong reasoning leading to wrong results, and (2) wrong reasoning that produces correct results. We illustrate both with the following examples.

### 6.1 Invalid Reasoning Leading to Incorrect Results

Consider the indefinite and definite integrals,

$$\int \frac{1 - \sin x}{x + \cos x}\, dx = \ln |x + \cos x| + C, \qquad \int_{-\pi/6}^{\pi/6} \frac{1 - \sin x}{x + \cos x}\, dx.$$

When asked for the corresponding *indefinite integral*, most models correctly applied the substitution $u = x + \cos x$, yielding the valid antiderivative $\ln |x + \cos x| + C$. However, when tasked with evaluating the *definite integral*, many models such as GPT-4.1 (see Solution 2) abandoned this approach. Instead, they applied a symmetry argument, incorrectly reasoning that the integrand was odd and the integral must vanish. In reality, the integrand is not odd, and the correct value is approximately $1.40$.

More broadly, LLMs often rely on shortcut strategies for prediction rather than carrying out careful justification (Yuan et al., 2024). Our example illustrates this shortcut issue in mathematics. Specifically, when faced with integrals under symmetric bounds, LLMs tend to rely on shortcut strategies

of symmetry-based reasoning rather than verifying conditions and executing systematic derivations. The same behavior extends across subjects of undergraduate mathematics, including misapplied algebraic identities, unjustified cancellations, and overgeneralization of familiar patterns. These errors indicate that current models are not failing at isolated techniques, but rather at the more complex task of reliably distinguishing between valid and invalid reasoning.

## 6.2 Invalid Reasoning Leading to Correct Results

For the same integral, some models, such as OpenAI-o3 and Mistral 7B Instruct, produced the correct numerical value, but through invalid reasoning (see Solution 3 and Solution 4). Instead of finding the antiderivative using traditional methods, they incorrectly claimed that no closed-form solution existed and switched to numerical approximation. OpenAI-o3 gave a value of $1.4511$, and Mistral 7B Instruct v0.3 gave $1.400731$, both close to the true result (approximately $1.40$) but achieved through flawed reasoning that created an illusion of success.

Such cases highlight a critical limitation of accuracy-based evaluation. When models arrive at correct answers through flawed reasoning, accuracy scores alone cannot reveal the underlying weaknesses. Similar patterns arise across undergraduate mathematics: models provide correct final results for limits, series, or differential equations while relying on deceptive arguments, unjustified approximations, or incomplete steps. Evaluations that stop at final-answer correctness, therefore, overestimate model competence. This underscores the need for frameworks that assess not only outcomes but also the validity and coherence of the reasoning process itself.

## 6.3 Implications

These two failure modes show a recurring pattern in LLM reasoning. Models often display local competence, solving individual steps correctly, but struggle to integrate them into globally consistent solutions. Their answers may look polished, but closer inspection shows reasoning that is weak, misleading, and unreliable. At the same time, these problems point to clear directions for improvement. Future models need better methods to maintain consistency, apply shortcuts carefully, and integrate symbolic reasoning with LLMs. Equally important, evaluation should extend beyond mere final-answer accuracy. Therefore, we need frameworks that assess the reasoning process, so that systems become not only fluent but also trustworthy mathematical problem solvers.

## 7 Conclusion

This paper introduces CUMath, a benchmark and evaluation framework designed to assess both correctness and reasoning quality in undergraduate-level computational mathematics. Our analysis reveals that even the strongest LLMs achieve an accuracy rate of less than 25%. While these models may occasionally provide correct answers, they frequently rely on flawed algebraic manipulations, misuse shortcuts, or exhibit inconsistent reasoning. These findings indicate that accuracy alone is an insufficient measure of mathematical competence. By combining symbolic verification, reasoning-sensitive metrics, and LLM-as-a-grader feedback, CUMath highlights weaknesses that traditional evaluation methods tend to overlook.

## Future Work

Our analysis indicates several potential directions for improvement. First, models should incorporate mechanisms that enforce global consistency, ensuring that locally correct steps lead to coherent solutions. Second, they require more selective use of simplifying strategies, applied only when assumptions are valid. Third, a closer integration of neural reasoning with symbolic tools could enhance reliability in tasks related to algebra and integration. Finally, evaluation should extend beyond final-answer accuracy. Metrics that capture correctness, coherence, and validity together will provide a more accurate measurement of mathematical competence and better inform future development.

ETHICS STATEMENT

CUMath is constructed from real instructional materials (university quizzes, exams, problem sets) and open-access textbooks (Appendix A). Textbook items are released under their respective Creative Commons licenses; closed instructional materials were included under explicit agreements with instructors. The dataset contains no personally identifiable information, human-subject data, or sensitive attributes; problems were reviewed by mathematics educators for clarity and curricular alignment.

Potential risks include (i) inadvertent training set overlap with future models and (ii) misuse of the benchmark or automatic grader for assessment without human oversight. To mitigate these risks, we (a) release provenance metadata and licensing information, (b) distribute CUMath for non-commercial research use, and (c) emphasize that the LLM-as-grader pipeline is for research evaluation—not a substitute for expert grading. We comply with the ICLR Code of Ethics and the legal terms of all sources, and we utilize Grammarly to enhance the paper's grammar and clarity.

REPRODUCIBILITY STATEMENT

We provide all resources needed to reproduce our results. The CUMath dataset and all prompt templates (Zero-Shot, CoT, ToT, SC, evaluation prompts) are included in Appendix D.2 and D.3 and released together with the code. The complete LLM-as-a-grader implementation, including both passes and verification loops, is part of the code release. Our evaluation pipeline specifies model names, decoding parameters (ToT: temperature $0.7$, 3 paths; SC: temperature $0.9$, 5 samples), maximum output length (2,048 tokens), and random seeds. Symbolic checks are performed with SymPy, external verification with the Wolfram Alpha Short Answers API, and path similarity with MathBERT. We release anonymized code, scripts, and configuration files to reproduce all reported tables and figures. Dataset source licenses are documented in the appendix.

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

## A  DATASET SOURCES

Table 3: Mapping of dataset domains to textbook sources and associated licenses.

| Domain | Textbook Source | Author(s) | License |
|---|---|---|---|
| Calculus | *APEX Calculus* | Gregory Hartman | CC BY 4.0 |
| Differential Equations | *Elementary Differential Equations (with BVP)* | William F. Trench | CC BY-SA 4.0 |
| Discrete Mathematics | *Discrete Mathematics: An Open Introduction* | Oscar Levin | CC BY-SA 4.0 |
| Linear Algebra | *A First Course in Linear Algebra* | Rob Beezer | CC BY-SA 4.0 |
| Multivariable Calculus | *APEX Calculus* | Gregory Hartman | CC BY 4.0 |
| Pre-calculus | *Precalculus / College Algebra / Trigonometry* | Carl Stitz, Jeff Zeager | CC BY-NC-SA 3.0 |
| Trigonometry | *Precalculus / College Algebra / Trigonometry* | Carl Stitz, Jeff Zeager | CC BY-NC-SA 3.0 |

CUMath also includes problems drawn from university-level quizzes and examinations authored by the course instructor. These materials are not publicly available; however, explicit licenses were obtained from the authors to incorporate them into our benchmark. We therefore designate these as *licensed instructor-authored problems*. Such items are tagged in the dataset metadata and are distributed only for non-commercial purposes.

Table 4: Source distribution by topic.

| Topic (file) | Total | Textbook | Real-world (from courses' problem sets, exams, and quizzes) | % Real World |
|---|---|---|---|---|
| Calculus | 300 | 21 | 279 | 93.0% |
| Differential Equations | 300 | 55 | 245 | 81.7% |
| Discrete Math | 300 | 154 | 146 | 48.7% |
| Linear Algebra | 300 | 97 | 203 | 67.7% |
| Multivariable Calculus | 300 | 52 | 248 | 82.7% |
| Pre-calculus | 300 | 114 | 186 | 62.0% |
| Trigonometry | 300 | 120 | 180 | 60.0% |
| **OVERALL** | 2100 | 613 | 1487 | 70.8% |

## B  SUB-TOPIC DISTRIBUTION

Table 5: Distribution of problems across sub-topics in **Calculus**.

| Sub-topic | Count |
|---|---|
| Definite Integral | 53 |
| Limit | 52 |
| Derivative | 44 |
| Indefinite Integral | 42 |
| Function Analysis | 30 |
| Sequence/Series | 30 |
| Real-world Problems (Optimization) | 21 |
| Continuity | 17 |
| Improper Integral | 11 |

Table 6: Distribution of problems across sub-topics in **Differential Equations**.

| Sub-topic | Count |
|---|---|
| Linear Second Order Equations | 135 |
| Laplace Transform | 56 |
| Linear First Order Equations | 37 |
| Exact Equations | 22 |
| Separable Equations | 17 |
| Transformation of Nonlinear Equations into Separable Equations | 20 |
| Existence and Uniqueness of Solutions of Nonlinear Equations | 13 |

Table 7: Distribution of problems across sub-topics in **Discrete Mathematics**.

| Sub-topic | Count |
|---|---|
| Sequences | 106 |
| Recurrences | 81 |
| Generating Functions | 34 |
| Number Theory | 32 |
| Sums & Products | 27 |
| Combinatorics | 18 |
| Logic | 2 |

Table 8: Distribution of problems across sub-topics in **Linear Algebra**.

| Sub-topic | Count |
|---|---|
| Linear Transformations | 45 |
| Linear Independence / Dependence | 44 |
| Eigenvalues, Eigenvectors & Characteristic Polynomial | 36 |
| Matrix Operations | 34 |
| Systems of Linear Equations | 31 |
| Spanning Sets, Rank & Dimension | 27 |
| Determinants | 23 |
| Matrix Properties & Operations | 14 |
| Vector Spaces / Subspaces | 14 |
| Vector Operations & Representations | 10 |
| Linear Transformations & Representations | 9 |
| Orthogonality / Inner Product | 8 |
| Null Space & Nullity | 5 |

Table 9: Distribution of problems across sub-topics in **Multivariable Calculus**.

| Sub-topic | Count |
|---|---|
| Vector Calculus | 80 |
| Multiple Integrals | 74 |
| Geometry of Space | 49 |
| Partial Derivatives | 46 |
| Limit | 20 |
| Function Analysis | 19 |
| Vector-valued Function | 6 |
| Real-world Problem (Optimization) | 6 |

Table 10: Distribution of problems across sub-topics in **Pre-calculus**.

| Sub-topic | Count |
|---|---|
| Functions | 162 |
| Applications | 46 |
| Equations | 34 |
| Polynomials | 33 |
| Log/Exponential | 13 |
| Inequalities | 12 |

Table 11: Distribution of problems across sub-topics in **Trigonometry**.

| Sub-topic | Count |
|---|---|
| Evaluating Trigonometric Functions | 70 |
| Inverse Trigonometric Functions | 62 |
| Trigonometric Equations | 49 |
| Exact Values | 39 |
| Trigonometric Identities | 30 |
| Solving Triangles | 19 |
| Angle Conversion | 16 |
| Unit Circle & Reference Angles | 12 |
| Trigonometric Inequalities | 3 |

## C  PROBLEM EXAMPLES

> **Example CUMath entry**
>
> ```
> "id":  "149",
> "topic":  "Single Variable Calculus",
> "subtopic":  "limit",
> "question":  Evaluate  lim  x^(sin x),
>                        x→0+
> "answer":  1,
> "steps":
>     • Let  y = x^(sin x).   Then  ln y = sin x · ln x.
>     • lim_(x→0+) ln y = lim_(x→0+) sin x · ln x
>     • Rewrite as a quotient:   lim_(x→0+) (ln x)/(1/sin x)
>     • Apply L'Hôpital's Rule:   lim_(x→0+) (sin x)²/(−x cos x) = 0
>     • So  lim_(x→0+) y = e⁰ = 1
> "source":  "Quizzes",
> "type":  "FR"
> ```

## D  DETAILED EXPERIMENTAL SETUP

### D.1  EVALUATED LLMS

Table 12: Detailed specifications of evaluated LLMs.

| Model | Type | Size | Release Date | Specialization |
|---|---|---|---|---|
| GPT-4.1 | Closed-source | Not disclosed | 2025 | General |
| GPT-3.5-turbo-0125 | Closed-source | ∼175B (est.) | 2024 | General |
| OpenAI o3 | Closed-source | Not disclosed | 2024 | General |
| Claude Sonnet 3.7 | Closed-source | Not disclosed | 2025 | General |
| DeepSeek-R1-Distill-Qwen-32B | Open-source | 32B (distilled) | 2025 | General |
| Gemma 2 9B IT | Open-source | 9B | 2024 | General |
| LLaMA 3 8B Instruct | Open-source | 8B | 2024 | General |
| LLaMA 3 70B Instruct | Open-source | 70B | 2024 | General |
| LLaMA 4 Scout 17B Instruct | Open-source | 17B | 2025 | General |
| Qwen2.5 7B Instruct | Open-source | 7B | 2025 | General |
| Mistral 7B Instruct v0.3 | Open-source | 7B | 2024 | General |
| Qwen2.5-Math-7B Instruct | Math-specialized | 7B | 2024 | Mathematical reasoning |
| Qwen2.5-Math-1.5B Instruct | Math-specialized | 1.5B | 2024 | Mathematical reasoning |
| Llemma-7B | Math-specialized | 7B | 2024 | Mathematical reasoning |
| LLaMA-3.2-1B Instruct (ai-nexuz ft.) | Math-specialized | 1B | 2024 | Mathematical reasoning |

## D.2 SOLUTION GENERATION PROMPTS

---

**Zero-Shot Prompt**

**System Prompt:** Conclude the final answer in the form:
`\boxed{your final answer here}.`
**User:** Solve the following math problem: {problem}

---

**Chain-of-Thoughts Prompt**

**System Prompt:** You are a highly skilled mathematics expert. Solve the problem step by step. Conclude with your final answer in the form:
`\boxed{your final answer here}.`
**User:** Q: {example-question-1}
A: {example-solution-steps}
Q: {example-question-2}
A: {example-solution-steps}
.
.
.
Q: {problem}
A:

---

**Tree-of-Thoughts Prompt**

**System:** You are a highly skilled mathematics expert. Brainstorm multiple distinct solution paths for the given problem. At the end, clearly state the final answer in the form:
`\boxed{your final answer here}.`
**User:** {problem}
A (Path 1): {reasoning}
A (Path 2): {reasoning}
... `\boxed{\{final-answer\}}`

---

**Self-Consistency Prompt**

**System:** You are a highly skilled mathematics expert. Solve the problem with clear, step-by-step reasoning. At the end, clearly state the final answer in the form:
`\boxed{your final answer here}.`
**User:** {problem}
A (Sample 1): {reasoning}
A (Sample 2): {reasoning}
...
`\boxed{\{final-answer\}}`

---

## D.3 EVALUATION PROMPTS (LLM-AS-A-GRADER)

---

**Pass 1 — Step Feedback + Score (No CAS)**

**System:** You are a meticulous and fair mathematics instructor.
Given a problem, its correct reference steps, and a proposed step-by-step solution, evaluate each proposed step *independently*. Score each step on a 1–5 scale (1=very poor, 5=excellent) based on: Correctness, Logic/Flow, Justification, and Clarity.
**Important:**
- Do *not* reference or claim any CAS results in this pass.
- If a step is prose (no explicit equality), still give feedback and a score.
- Judge each step as written; do not merge or rewrite steps.

---

**Problem:** {problem}
**Reference Solution Steps:**
Ref Step 1: {ref_step_1}
Ref Step 2: {ref_step_2}
. . .
**Proposed Solution Steps:**
Step 1: {model_step_1}
Step 2: {model_step_2}
. . .

---

**Respond EXACTLY in this format (one line per student step):**
Step 1: [1–2 sentences of feedback]   Score: [X]/5
Step 2: [1–2 sentences of feedback]   Score: [Y]/5
. . .

Evaluation Prompt — Pass 2 (Step Feedback + Score)

**System:** You revise scores for math steps using CAS results.
If CAS shows an incorrect transformation, lower the score; if it confirms, consider raising.
If both CAS statuses are 'unknown', keep the score unchanged and note 'CAS unknown'.
Return STRICT JSON list:
```
[{ "idx": int, "revised": int (1..5), "note": str }].
```

### D.4 DETAILED COMPUTATION OF STEPWISE REASONING SCORE

We follow the ROSCOE framework (Golovneva et al., 2023) to evaluate the quality of reasoning chains produced by $M$. This section provides the exact computation of the six metrics we use: Faithfulness, Informativeness (Step), Informativeness (Chain), Repetition (Step), Discourse Representation, and Coherence (Step vs. Step). We implement faithfulness/informativeness via token/sentence–step cosine alignment and use an NLI model to penalize contradictions for Discourse/Coherence.

We represent each problem statement $q_i$ as a sequence of tokens: $q_i = \{q_{i,1}, q_{i,2}, \ldots, q_{i,|q_i|}\}$, where $q_{i,t}$ denotes the embedding of the $t$-th token in $q_i$.

**Faithfulness** $(e_i \to q_i)$. Measures whether each generated step is grounded in the problem statement:

$$\text{Faithfulness}(e_i) = \frac{1}{\hat{n}_i} \sum_{j=1}^{\hat{n}_i} r\text{-align}(e_i^j \to q_i), \qquad r\text{-align}(e_i^j \to q_i) = \frac{1 + \max_{t=1..|q_i|} \cos(e_i^j, q_{i,t})}{2}.$$

**Informativeness (Step)** $(e_i \leftrightarrow q_i)$. Captures how well information in the problem statement is reflected in the generated reasoning:

$$\text{Info-Step}(e_i) = \frac{1}{2} \left( \frac{1}{|q_i|} \sum_{t=1}^{|q_i|} r\text{-align}(q_{i,t} \to e_i) + \frac{1}{\hat{n}_i} \sum_{j=1}^{\hat{n}_i} r\text{-align}(e_i^j \to q_i) \right).$$

**Informativeness (Chain)** $(e_i \implies q_i)$. Measures agreement between the reasoning chain and the problem statement as a whole:

$$\text{Info-Chain}(e_i) = \frac{1 + \cos(e_i, q_i)}{2}.$$

**Repetition (Step)** $(\hat{s}_i^j \leftrightarrow \hat{s}_i^k)$. To identify repeated or paraphrased reasoning steps, we measure similarity between embeddings of different steps in the reasoning chain. Each step $\hat{s}_i^j$ is represented as a single embedding, and repetition is computed via cosine similarity between step embeddings:

$$\text{Repetition-Step}(e_i) = \frac{1 - \max_{j=2..\hat{n}_i} \max_{k=1..j-1} \cos(\hat{s}_i^j, \hat{s}_i^k)}{2}.$$

**Discourse Representation** $(e_i \Leftrightarrow q_i)$. Assesses whether any generated step contradicts the problem statement:

$$\text{Discourse}(e_i) = 1 - \max_{j=1..\hat{n}_i,\, t=1..|q_i|} p_{\text{contr}}(\hat{s}_i^j, q_{i,t}),$$

where $p_{\text{contr}}$ is the contradiction probability predicted by a natural language inference (NLI) model.

**Coherence (Step vs. Step)**. Checks for contradictions between generated steps:

$$\text{Coherence}(e_i) = 1 - \max_{j=2..\hat{n}_i,\, k<j} p_{\text{contr}}(\hat{s}_i^j, \hat{s}_i^k).$$

# E    DETAILED RESULTS

Table 13: **Main Results on Calculus.** Evaluation of LLMs across four prompting strategies and five metrics: Accuracy (Acc), Semantic F1, SRS, VR, and LLM-based evaluation (LLM, normalized to [0,1]). The highest value in each column is **bold and underlined**.

| Model | Zero-shot | | | | | CoT | | | | | ToT | | | | | SC | | | | |
|---|---|---|---|---|---|---|---|---|---|---|---|---|---|---|---|---|---|---|---|---|
| | Acc | F1 | SRS | VR | LLM | Acc | F1 | SRS | VR | LLM | Acc | F1 | SRS | VR | LLM | Acc | F1 | SRS | VR | LLM |
| *Closed-source Models* | | | | | | | | | | | | | | | | | | | | |
| GPT-4.1 | 0.24 | 0.22 | 0.35 | -0.93 | 0.53 | 0.12 | 0.09 | 0.38 | -0.61 | 0.56 | 0.24 | 0.05 | 0.35 | -0.92 | 0.27 | 0.12 | 0.02 | 0.37 | -0.77 | 0.25 |
| GPT-3.5-turbo-0125 | 0.24 | 0.15 | **0.42** | -0.51 | 0.52 | 0.08 | **0.24** | 0.43 | -0.65 | 0.55 | 0.26 | 0.16 | 0.37 | -0.95 | 0.57 | 0.07 | 0.08 | 0.39 | -0.87 | 0.58 |
| OpenAI o3 | 0.25 | 0.24 | 0.39 | -0.64 | 0.67 | 0.11 | 0.14 | **0.45** | **-0.33** | 0.69 | 0.21 | 0.07 | 0.36 | -0.92 | **0.60** | 0.11 | 0.05 | **0.41** | -0.53 | 0.58 |
| Claude Sonnet 3.7 | 0.26 | 0.29 | 0.40 | -0.53 | **0.73** | 0.11 | 0.19 | 0.41 | -0.71 | **0.73** | 0.24 | 0.15 | **0.40** | -0.83 | 0.59 | 0.11 | **0.16** | **0.41** | -0.75 | **0.67** |
| *Open-source Models* | | | | | | | | | | | | | | | | | | | | |
| DeepSeek-R1-Distill-Qwen-32B | **0.29** | 0.17 | 0.38 | -0.95 | 0.67 | 0.12 | 0.08 | 0.40 | -0.72 | 0.67 | **0.32** | 0.06 | 0.38 | -0.96 | 0.37 | 0.12 | 0.02 | 0.37 | -0.92 | 0.37 |
| Gemma 2 9B IT | 0.18 | 0.10 | 0.39 | -0.65 | 0.53 | 0.09 | 0.12 | 0.42 | -0.49 | 0.55 | 0.21 | 0.13 | 0.37 | -0.94 | 0.22 | 0.03 | 0.04 | 0.39 | -0.74 | 0.22 |
| LLaMA 3 8B Instruct | 0.21 | 0.08 | 0.41 | -0.77 | 0.62 | 0.08 | 0.15 | 0.41 | -0.68 | 0.48 | 0.21 | 0.08 | 0.39 | -0.94 | 0.30 | 0.06 | 0.04 | 0.39 | -0.91 | 0.38 |
| LLaMA 3 70B Instruct | 0.26 | **0.40** | 0.39 | -0.52 | 0.67 | 0.09 | 0.22 | 0.41 | -0.66 | 0.67 | 0.29 | 0.12 | 0.37 | -0.70 | 0.26 | 0.06 | 0.08 | 0.38 | -0.90 | 0.27 |
| LLaMA 4 Scout 17B Instruct | 0.26 | 0.18 | 0.40 | -0.77 | 0.67 | **0.13** | 0.18 | 0.39 | -0.75 | 0.65 | 0.26 | **0.20** | 0.38 | -0.98 | 0.44 | **0.13** | 0.05 | 0.38 | -0.91 | 0.37 |
| Qwen2.5 7B Instruct | 0.26 | 0.14 | 0.35 | -0.89 | 0.55 | 0.12 | 0.12 | 0.39 | -0.69 | 0.56 | 0.11 | 0.03 | 0.36 | -0.81 | 0.30 | 0.11 | 0.03 | 0.37 | -0.87 | 0.38 |
| Mistral 7B Instruct v0.3 | 0.09 | 0.29 | 0.38 | **-0.15** | 0.29 | 0.04 | 0.15 | 0.43 | -0.62 | 0.40 | 0.21 | 0.06 | 0.38 | -0.92 | 0.37 | 0.01 | 0.05 | 0.39 | -0.89 | 0.40 |
| *Math-specialized Models* | | | | | | | | | | | | | | | | | | | | |
| Qwen2.5-Math-7B Instruct | 0.15 | 0.24 | 0.35 | -0.93 | 0.47 | 0.05 | 0.13 | 0.38 | -0.71 | 0.29 | 0.07 | 0.03 | 0.35 | -0.86 | 0.32 | 0.06 | 0.02 | 0.34 | -0.90 | 0.23 |
| Llemma-7B | 0.00 | 0.03 | 0.34 | -0.53 | 0.23 | 0.01 | 0.02 | 0.39 | -0.44 | 0.28 | 0.01 | 0.01 | 0.36 | -0.83 | 0.09 | 0.01 | 0.01 | 0.33 | -0.95 | 0.11 |
| Qwen2.5-Math-1.5B Instruct | 0.09 | 0.36 | 0.36 | -0.92 | 0.39 | 0.06 | 0.10 | 0.38 | -0.66 | 0.49 | 0.07 | 0.02 | 0.38 | -0.40 | 0.17 | 0.03 | 0.01 | 0.38 | **-0.38** | 0.13 |
| LLaMA-3.2-1B Instruct (ft) | 0.17 | 0.38 | 0.38 | -0.67 | 0.48 | 0.03 | 0.11 | 0.42 | -0.62 | 0.33 | 0.01 | 0.02 | 0.38 | **-0.30** | 0.16 | 0.01 | 0.01 | 0.38 | -0.40 | 0.13 |

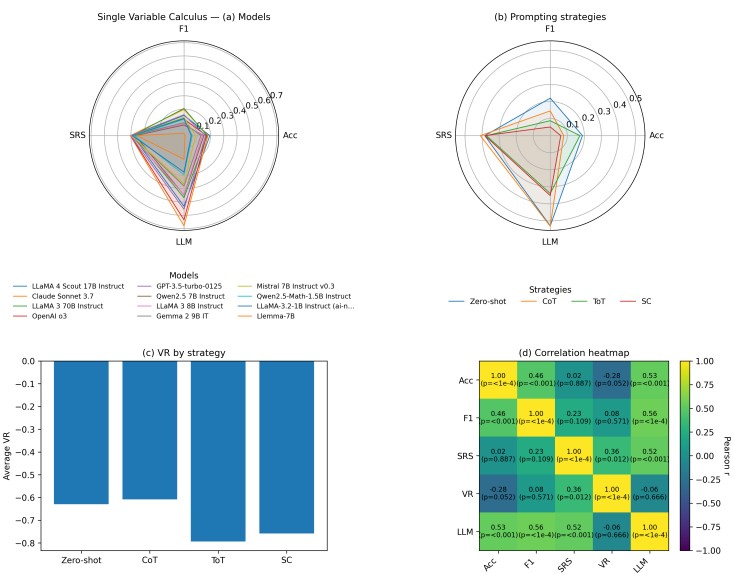

Figure 4: **Multi-metric summary of Calculus results.** (a) Average scores across models. (b) Average scores across prompting methods. (c) Average VR across prompting methods. (d) Pearson correlation heatmap with $p-$values.

Table 14: **Main Results on Differential Equations.** Evaluation of LLMs across four prompting strategies and five metrics: Accuracy (Acc), Semantic F1, SRS, VR, and LLM-based evaluation (LLM, normalized to [0,1]). The highest value in each column is **bold and underlined**.

| Model | Zero-shot | | | | | CoT | | | | | ToT | | | | | SC | | | | |
|---|---|---|---|---|---|---|---|---|---|---|---|---|---|---|---|---|---|---|---|---|
| | Acc | F1 | SRS | VR | LLM | Acc | F1 | SRS | VR | LLM | Acc | F1 | SRS | VR | LLM | Acc | F1 | SRS | VR | LLM |
| *Closed-source Models* | | | | | | | | | | | | | | | | | | | | |
| GPT-4.1 | 0.08 | 0.08 | 0.37 | -0.55 | 0.53 | 0.09 | 0.09 | 0.37 | -0.53 | 0.56 | 0.07 | 0.03 | 0.36 | -0.71 | 0.27 | **0.10** | 0.02 | 0.36 | -0.74 | 0.25 |
| GPT-3.5-turbo-0125 | 0.05 | **0.22** | 0.42 | -0.50 | 0.52 | 0.06 | **0.30** | 0.42 | -0.45 | 0.55 | 0.04 | 0.10 | 0.39 | -0.74 | 0.57 | 0.05 | 0.09 | 0.39 | -0.83 | 0.58 |
| OpenAI o3 | **0.09** | 0.16 | **0.46** | **-0.20** | 0.67 | **0.10** | 0.13 | **0.46** | **-0.24** | 0.69 | **0.08** | 0.05 | **0.45** | **-0.27** | **0.60** | 0.09 | 0.06 | **0.41** | **-0.38** | 0.58 |
| Claude Sonnet 3.7 | 0.07 | 0.18 | 0.40 | -0.71 | **0.73** | 0.08 | 0.17 | 0.40 | -0.64 | **0.73** | 0.07 | **0.12** | 0.39 | -0.74 | 0.59 | 0.07 | **0.17** | 0.40 | -0.68 | **0.67** |
| *Open-source Models* | | | | | | | | | | | | | | | | | | | | |
| DeepSeek-R1-Distill-Qwen-32B | 0.05 | 0.09 | 0.39 | -0.65 | 0.63 | 0.06 | 0.09 | 0.38 | -0.67 | 0.55 | 0.04 | 0.04 | 0.37 | -0.86 | 0.31 | 0.05 | 0.02 | 0.37 | -0.90 | 0.21 |
| Gemma 2 9B IT | 0.04 | 0.13 | 0.41 | -0.45 | 0.61 | 0.05 | 0.18 | 0.42 | -0.44 | 0.63 | 0.03 | 0.07 | 0.38 | -0.67 | 0.54 | 0.04 | 0.05 | 0.38 | -0.73 | 0.40 |
| LLaMA 3 8B Instruct | 0.04 | 0.14 | 0.40 | -0.62 | 0.62 | 0.05 | 0.20 | 0.40 | -0.60 | 0.64 | 0.04 | 0.05 | 0.38 | -0.85 | 0.52 | 0.05 | 0.04 | 0.38 | -0.90 | 0.56 |
| LLaMA 3 70B Instruct | 0.08 | 0.18 | 0.45 | -0.37 | 0.41 | 0.09 | **0.30** | 0.42 | -0.47 | 0.61 | 0.07 | 0.09 | 0.38 | -0.81 | 0.57 | 0.08 | 0.09 | 0.38 | -0.84 | 0.55 |
| LLaMA 4 Scout 17B Instruct | 0.05 | 0.17 | 0.39 | -0.71 | 0.67 | 0.06 | 0.22 | 0.39 | -0.63 | 0.65 | 0.04 | 0.09 | 0.37 | -0.85 | 0.44 | 0.05 | 0.05 | 0.37 | -0.90 | 0.37 |
| Qwen2.5 7B Instruct | 0.04 | 0.12 | 0.38 | -0.62 | 0.55 | 0.05 | 0.14 | 0.39 | -0.57 | 0.56 | 0.03 | 0.04 | 0.36 | -0.79 | 0.30 | 0.04 | 0.03 | 0.37 | -0.82 | 0.38 |
| Mistral 7B Instruct v0.3 | 0.03 | 0.10 | 0.43 | -0.47 | 0.29 | 0.04 | 0.20 | 0.42 | -0.48 | 0.40 | 0.03 | 0.05 | 0.38 | -0.82 | 0.37 | 0.04 | 0.04 | 0.39 | -0.87 | 0.40 |
| *Math-specialized Models* | | | | | | | | | | | | | | | | | | | | |
| Qwen2.5-Math-7B Instruct | 0.02 | 0.10 | 0.38 | -0.62 | 0.47 | 0.03 | 0.05 | 0.39 | -0.49 | 0.29 | 0.02 | 0.03 | 0.34 | -0.79 | 0.32 | 0.03 | 0.02 | 0.34 | -0.83 | 0.23 |
| Llemma-7B | 0.01 | 0.03 | 0.40 | -0.38 | 0.23 | 0.02 | 0.03 | 0.38 | -0.41 | 0.28 | 0.01 | 0.01 | 0.35 | -0.80 | 0.09 | 0.01 | 0.01 | 0.33 | -0.94 | 0.11 |
| Qwen2.5-Math-1.5B Instruct | 0.02 | 0.08 | 0.39 | -0.56 | 0.39 | 0.02 | 0.09 | 0.38 | -0.58 | 0.49 | 0.01 | 0.04 | 0.37 | -0.74 | 0.17 | 0.01 | 0.03 | 0.37 | -0.79 | 0.13 |
| LLaMA-3.2-1B Instruct (ft) | 0.02 | 0.05 | 0.41 | -0.50 | 0.48 | 0.02 | 0.06 | 0.41 | -0.51 | 0.33 | 0.01 | 0.03 | 0.39 | -0.76 | 0.16 | 0.01 | 0.02 | 0.38 | -0.83 | 0.13 |

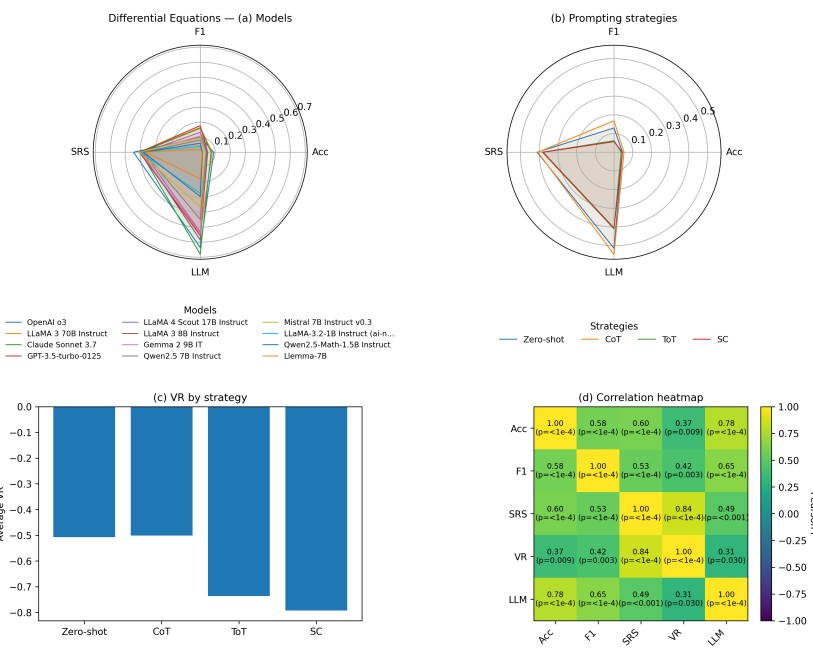

Figure 5: **Multi-metric summary of Differential Equations results.** (a) Average scores across models. (b) Average scores across prompting methods. (c) Average VR across prompting methods. (d) Pearson correlation heatmap with $p-$values.

Table 15: **Main Results on Discrete Mathematics.** Evaluation of LLMs across four prompting strategies and five metrics: Accuracy (Acc), Semantic F1, SRS, VR, and LLM-based evaluation (LLM, normalized to [0,1]). The highest value in each column is **bold and underlined**.

| Model | Zero-shot | | | | | CoT | | | | | ToT | | | | | SC | | | | |
|---|---|---|---|---|---|---|---|---|---|---|---|---|---|---|---|---|---|---|---|---|
| | Acc | F1 | SRS | VR | LLM | Acc | F1 | SRS | VR | LLM | Acc | F1 | SRS | VR | LLM | Acc | F1 | SRS | VR | LLM |
| *Closed-source Models* | | | | | | | | | | | | | | | | | | | | |
| GPT-4.1 | 0.17 | 0.08 | 0.40 | -0.66 | 0.60 | 0.15 | 0.09 | 0.41 | -0.63 | 0.59 | 0.16 | 0.02 | 0.37 | -0.87 | 0.26 | 0.16 | 0.02 | 0.37 | -0.85 | 0.26 |
| GPT-3.5-turbo-0125 | 0.17 | **0.15** | 0.50 | -0.49 | 0.51 | 0.16 | **0.18** | **0.50** | -0.46 | 0.53 | 0.16 | **0.05** | 0.40 | -0.83 | 0.53 | 0.13 | 0.05 | 0.41 | -0.86 | 0.51 |
| OpenAI o3 | 0.11 | 0.13 | 0.52 | **-0.38** | 0.65 | 0.11 | 0.10 | 0.48 | -0.47 | **0.73** | 0.11 | 0.03 | **0.41** | **-0.71** | 0.51 | 0.12 | 0.04 | 0.41 | **-0.71** | 0.57 |
| Claude Sonnet 3.7 | 0.16 | 0.10 | 0.43 | -0.73 | **0.72** | **0.24** | 0.10 | 0.43 | -0.70 | 0.71 | 0.16 | **0.05** | **0.41** | -0.78 | **0.57** | 0.15 | **0.07** | **0.43** | -0.75 | **0.63** |
| *Open-source Models* | | | | | | | | | | | | | | | | | | | | |
| DeepSeek-R1-Distill-Qwen-32B | 0.15 | 0.06 | 0.40 | -0.87 | 0.55 | 0.16 | 0.06 | 0.39 | -0.88 | 0.51 | 0.15 | 0.02 | 0.38 | -0.96 | 0.27 | 0.16 | 0.01 | 0.38 | -0.96 | 0.92 |
| Gemma 2 9B IT | 0.15 | 0.10 | 0.46 | -0.58 | 0.56 | 0.16 | 0.13 | **0.50** | -0.48 | 0.55 | 0.11 | 0.04 | **0.41** | -0.76 | 0.47 | 0.10 | 0.03 | 0.41 | -0.77 | 0.39 |
| LLaMA 3 8B Instruct | 0.14 | 0.09 | 0.44 | -0.71 | 0.61 | 0.16 | 0.11 | 0.43 | -0.73 | 0.64 | **0.23** | 0.03 | 0.40 | -0.87 | 0.43 | 0.11 | 0.03 | 0.40 | -0.91 | 0.44 |
| LLaMA 3 70B Instruct | 0.18 | 0.14 | **0.53** | -0.41 | 0.50 | 0.18 | 0.14 | 0.46 | -0.58 | 0.64 | 0.19 | 0.04 | 0.40 | -0.84 | 0.46 | 0.16 | 0.04 | 0.40 | -0.86 | 0.49 |
| LLaMA 4 Scout 17B Instruct | **0.21** | 0.10 | 0.42 | -0.82 | 0.69 | 0.21 | 0.11 | 0.43 | -0.79 | 0.65 | 0.19 | 0.04 | 0.40 | -0.88 | 0.40 | **0.20** | 0.03 | 0.40 | -0.91 | 0.33 |
| Qwen2.5 7B Instruct | 0.17 | 0.09 | 0.40 | -0.76 | 0.63 | 0.18 | 0.11 | 0.40 | -0.74 | 0.63 | 0.17 | 0.03 | 0.37 | -0.89 | 0.38 | 0.17 | 0.02 | 0.38 | -0.91 | 0.39 |
| Mistral 7B Instruct v0.3 | 0.08 | 0.09 | 0.51 | -0.42 | 0.37 | 0.05 | 0.14 | 0.43 | **-0.44** | 0.39 | 0.10 | 0.04 | **0.41** | -0.86 | 0.39 | 0.05 | 0.03 | 0.40 | -0.87 | 0.38 |
| *Math-specialized Models* | | | | | | | | | | | | | | | | | | | | |
| Qwen2.5-Math-7B Instruct | 0.07 | 0.08 | 0.39 | -0.79 | 0.55 | 0.05 | 0.02 | 0.40 | -0.61 | 0.26 | 0.06 | 0.02 | 0.35 | -0.90 | 0.26 | 0.06 | 0.01 | 0.35 | -0.93 | 0.48 |
| Llemma-7B | 0.00 | 0.01 | 0.43 | -0.49 | 0.26 | 0.02 | 0.02 | 0.42 | -0.45 | 0.25 | 0.01 | 0.00 | 0.38 | -0.86 | 0.41 | 0.00 | 0.00 | 0.34 | -0.96 | 0.41 |
| Qwen2.5-Math-1.5B Instruct | 0.01 | 0.04 | 0.45 | -0.40 | 0.41 | 0.05 | 0.06 | 0.39 | -0.77 | 0.50 | 0.07 | 0.02 | 0.38 | -0.86 | 0.32 | 0.06 | 0.01 | 0.37 | -0.91 | 0.49 |
| LLaMA-3.2-1B Instruct (ft) | 0.02 | 0.06 | 0.44 | -0.68 | 0.45 | 0.01 | 0.07 | 0.44 | -0.67 | 0.48 | 0.05 | 0.03 | 0.40 | -0.84 | 0.32 | 0.04 | 0.02 | 0.40 | -0.88 | 0.30 |

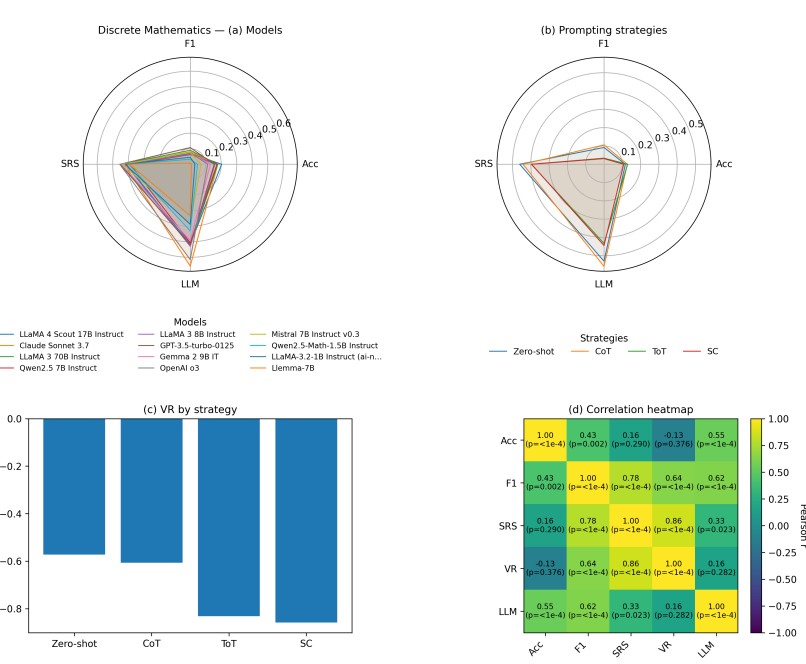

Figure 6: **Multi-metric summary of Discrete Mathematics results.** (a) Average scores across models. (b) Average scores across prompting methods. (c) Average VR across prompting methods. (d) Pearson correlation heatmap with $p-$values.

Table 16: **Main Results on Multivariable Calculus.** Evaluation of LLMs across four prompting strategies and five metrics: Accuracy (Acc), Semantic F1, SRS, VR, and LLM-based evaluation (LLM, normalized to [0,1]). The highest value in each column is **bold and underlined**.

| Model | Zero-shot | | | | | CoT | | | | | ToT | | | | | SC | | | | |
|---|---|---|---|---|---|---|---|---|---|---|---|---|---|---|---|---|---|---|---|---|
| | Acc | F1 | SRS | VR | LLM | Acc | F1 | SRS | VR | LLM | Acc | F1 | SRS | VR | LLM | Acc | F1 | SRS | VR | LLM |
| *Closed-source Models* | | | | | | | | | | | | | | | | | | | | |
| GPT-4.1 | 0.07 | 0.06 | 0.53 | 0.10 | 0.53 | 0.07 | 0.06 | 0.57 | 0.10 | 0.56 | 0.07 | 0.01 | 0.76 | 0.02 | 0.27 | 0.08 | 0.01 | 0.73 | 0.02 | 0.25 |
| GPT-3.5-turbo-0125 | 0.06 | 0.15 | 0.39 | **0.20** | 0.52 | 0.03 | 0.12 | 0.38 | **0.17** | 0.55 | 0.02 | **0.03** | 0.51 | 0.05 | 0.57 | 0.03 | 0.02 | 0.60 | 0.05 | 0.58 |
| OpenAI o3 | 0.08 | **0.24** | 0.23 | 0.10 | 0.67 | 0.07 | **0.23** | 0.30 | 0.10 | 0.69 | 0.08 | 0.01 | 0.46 | 0.03 | **0.60** | 0.08 | 0.02 | 0.50 | 0.03 | 0.58 |
| Claude Sonnet 3.7 | 0.08 | 0.07 | **0.54** | 0.12 | **0.73** | 0.05 | 0.06 | **0.55** | 0.10 | **0.73** | 0.09 | **0.03** | **0.61** | **0.06** | 0.59 | 0.08 | **0.05** | 0.59 | **0.09** | **0.67** |
| *Open-source Models* | | | | | | | | | | | | | | | | | | | | |
| DeepSeek-R1-Distill-Qwen-32B | 0.08 | 0.03 | 0.58 | 0.06 | 0.67 | 0.07 | 0.03 | 0.54 | 0.06 | 0.67 | 0.07 | 0.01 | 0.70 | 0.02 | 0.37 | 0.07 | 0.01 | 0.75 | 0.01 | 0.37 |
| Gemma 2 9B IT | 0.04 | 0.07 | 0.34 | 0.10 | 0.53 | 0.03 | 0.06 | 0.28 | 0.09 | 0.55 | 0.04 | 0.02 | 0.51 | 0.04 | 0.22 | 0.01 | 0.01 | 0.57 | 0.03 | 0.22 |
| LLaMA 3 8B Instruct | 0.05 | 0.07 | 0.41 | 0.11 | 0.62 | 0.03 | 0.05 | 0.39 | 0.09 | 0.48 | 0.02 | 0.01 | 0.57 | 0.03 | 0.30 | 0.04 | 0.01 | 0.65 | 0.02 | 0.38 |
| LLaMA 3 70B Instruct | 0.09 | 0.09 | 0.23 | 0.12 | 0.67 | 0.04 | 0.07 | 0.38 | 0.12 | 0.67 | 0.04 | 0.02 | 0.56 | 0.04 | 0.26 | 0.03 | 0.02 | 0.64 | 0.04 | 0.27 |
| LLaMA 4 Scout 17B Instruct | **0.13** | 0.06 | 0.49 | 0.11 | 0.67 | **0.11** | 0.06 | 0.49 | 0.11 | 0.65 | **0.10** | 0.02 | 0.57 | 0.04 | 0.44 | **0.13** | 0.01 | 0.64 | 0.03 | 0.37 |
| Qwen2.5 7B Instruct | 0.10 | 0.07 | 0.51 | 0.11 | 0.55 | 0.08 | 0.04 | **0.55** | 0.07 | 0.56 | 0.09 | 0.01 | **0.61** | 0.02 | 0.30 | 0.07 | 0.01 | **0.68** | 0.02 | 0.38 |
| Mistral 7B Instruct v0.3 | 0.03 | 0.12 | 0.28 | 0.15 | 0.29 | 0.01 | 0.08 | 0.32 | 0.12 | 0.40 | 0.00 | 0.02 | 0.46 | 0.03 | 0.37 | 0.01 | 0.01 | 0.50 | 0.03 | 0.40 |
| *Math-specialized Models* | | | | | | | | | | | | | | | | | | | | |
| Qwen2.5-Math-7B Instruct | 0.01 | 0.03 | 0.47 | 0.06 | 0.47 | 0.01 | 0.03 | 0.41 | 0.05 | 0.29 | 0.02 | 0.01 | 0.57 | 0.02 | 0.32 | 0.01 | 0.01 | 0.60 | 0.01 | 0.23 |
| Llemma-7B | 0.01 | 0.00 | 0.15 | 0.01 | 0.23 | 0.01 | 0.00 | 0.15 | 0.01 | 0.28 | 0.00 | 0.00 | 0.35 | 0.00 | 0.09 | 0.00 | 0.00 | 0.45 | 0.00 | 0.11 |
| Qwen2.5-Math-1.5B Instruct | 0.04 | 0.02 | 0.42 | 0.04 | 0.39 | 0.03 | 0.02 | 0.39 | 0.04 | 0.49 | 0.05 | 0.01 | 0.55 | 0.02 | 0.17 | 0.04 | 0.01 | 0.59 | 0.01 | 0.13 |
| LLaMA-3.2-1B Instruct (ft) | 0.01 | 0.04 | 0.29 | 0.06 | 0.48 | 0.01 | 0.04 | 0.27 | 0.06 | 0.33 | 0.01 | 0.01 | 0.47 | 0.02 | 0.16 | 0.01 | 0.01 | 0.47 | 0.01 | 0.13 |

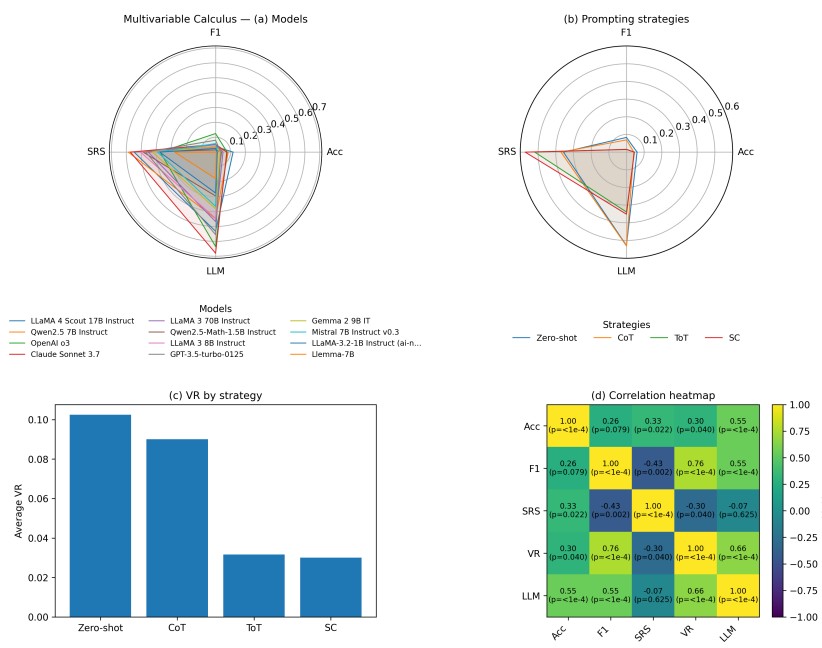

Figure 7: **Multi-metric summary of Multivariable Calculus results.** (a) Average scores across models. (b) Average scores across prompting methods. (c) Average VR across prompting methods. (d) Pearson correlation heatmap with $p-$values.

Table 17: **Main Results on Linear Algebra.** Evaluation of LLMs across four prompting strategies and five metrics: Accuracy (Acc), Semantic F1, SRS, VR, and LLM-based evaluation (LLM, normalized to [0,1]). The highest value in each column is **bold and underlined**.

| Model | Zero-shot | | | | | CoT | | | | | ToT | | | | | SC | | | | |
|---|---|---|---|---|---|---|---|---|---|---|---|---|---|---|---|---|---|---|---|---|
| | Acc | F1 | SRS | VR | LLM | Acc | F1 | SRS | VR | LLM | Acc | F1 | SRS | VR | LLM | Acc | F1 | SRS | VR | LLM |
| *Closed-source Models* | | | | | | | | | | | | | | | | | | | | |
| GPT-4.1 | 0.09 | 0.06 | 0.38 | -0.43 | 0.53 | 0.12 | 0.06 | 0.38 | -0.44 | 0.56 | 0.08 | 0.02 | 0.37 | -0.64 | 0.27 | 0.10 | 0.01 | 0.37 | -0.63 | 0.25 |
| GPT-3.5-turbo-0125 | 0.10 | **0.20** | 0.43 | -0.39 | 0.52 | 0.11 | **0.28** | 0.44 | -0.38 | 0.55 | 0.11 | **0.07** | 0.39 | -0.72 | 0.57 | 0.11 | 0.06 | 0.39 | -0.76 | 0.58 |
| OpenAI o3 | 0.08 | 0.07 | **0.47** | -0.06 | 0.67 | 0.08 | 0.05 | **0.45** | **-0.08** | 0.69 | 0.07 | 0.02 | **0.43** | **-0.24** | **0.60** | 0.08 | 0.03 | **0.41** | **-0.23** | 0.58 |
| Claude Sonnet 3.7 | 0.09 | 0.11 | 0.41 | -0.52 | **0.73** | 0.09 | 0.11 | 0.41 | -0.52 | **0.73** | 0.08 | 0.06 | 0.40 | -0.54 | 0.59 | 0.09 | **0.09** | **0.41** | -0.50 | **0.67** |
| *Open-source Models* | | | | | | | | | | | | | | | | | | | | |
| DeepSeek-R1-Distill-Qwen-32B | 0.05 | 0.04 | 0.39 | -0.57 | 0.67 | 0.11 | 0.04 | 0.39 | -0.57 | 0.67 | 0.06 | 0.02 | 0.38 | -0.77 | 0.37 | 0.10 | 0.01 | 0.37 | -0.82 | 0.37 |
| Gemma 2 9B IT | 0.10 | 0.10 | 0.42 | -0.34 | 0.53 | 0.14 | 0.18 | 0.44 | -0.30 | 0.55 | 0.08 | 0.04 | 0.40 | -0.55 | 0.22 | 0.05 | 0.04 | 0.39 | -0.62 | 0.22 |
| LLaMA 3 8B Instruct | 0.19 | 0.13 | 0.42 | -0.46 | 0.62 | 0.20 | 0.19 | 0.41 | -0.50 | 0.48 | 0.19 | 0.04 | 0.39 | -0.68 | 0.30 | **0.21** | 0.03 | 0.38 | -0.76 | 0.38 |
| LLaMA 3 70B Instruct | 0.12 | 0.17 | 0.44 | -0.37 | 0.67 | 0.15 | 0.26 | 0.44 | -0.43 | 0.67 | 0.11 | 0.05 | 0.39 | -0.75 | 0.26 | 0.10 | 0.05 | 0.39 | -0.72 | 0.27 |
| LLaMA 4 Scout 17B Instruct | **0.21** | 0.10 | 0.40 | -0.57 | 0.67 | **0.22** | 0.14 | 0.40 | -0.53 | 0.65 | **0.21** | 0.04 | 0.38 | -0.71 | 0.44 | **0.21** | 0.03 | 0.38 | -0.74 | 0.37 |
| Qwen2.5 7B Instruct | 0.09 | 0.10 | 0.39 | -0.56 | 0.55 | 0.12 | 0.10 | 0.39 | -0.56 | 0.56 | 0.09 | 0.03 | 0.37 | -0.71 | 0.30 | 0.10 | 0.03 | 0.37 | -0.75 | 0.38 |
| Mistral 7B Instruct v0.3 | 0.10 | 0.12 | 0.43 | -0.38 | 0.29 | 0.15 | 0.22 | **0.45** | -0.34 | 0.40 | 0.06 | 0.05 | 0.39 | -0.73 | 0.37 | 0.08 | 0.04 | 0.39 | -0.78 | 0.40 |
| *Math-specialized Models* | | | | | | | | | | | | | | | | | | | | |
| Qwen2.5-Math-7B Instruct | 0.02 | 0.07 | 0.38 | -0.62 | 0.47 | 0.04 | 0.04 | 0.41 | -0.46 | 0.29 | 0.03 | 0.02 | 0.34 | -0.80 | 0.32 | 0.02 | 0.01 | 0.34 | -0.84 | 0.23 |
| Llemma-7B | 0.01 | 0.03 | 0.39 | -0.34 | 0.23 | 0.03 | 0.03 | 0.39 | -0.35 | 0.28 | 0.01 | 0.01 | 0.36 | -0.69 | 0.09 | 0.01 | 0.01 | 0.34 | -0.85 | 0.11 |
| Qwen2.5-Math-1.5B Instruct | 0.01 | 0.02 | 0.46 | **-0.04** | 0.39 | 0.01 | 0.04 | 0.41 | -0.36 | 0.49 | 0.02 | 0.02 | 0.37 | -0.71 | 0.17 | 0.02 | 0.01 | 0.36 | -0.76 | 0.13 |
| LLaMA-3.2-1B Instruct (ft) | 0.01 | 0.10 | 0.42 | -0.41 | 0.48 | 0.05 | 0.08 | 0.40 | -0.49 | 0.33 | 0.05 | 0.04 | 0.39 | -0.66 | 0.16 | 0.04 | 0.03 | 0.38 | -0.75 | 0.13 |

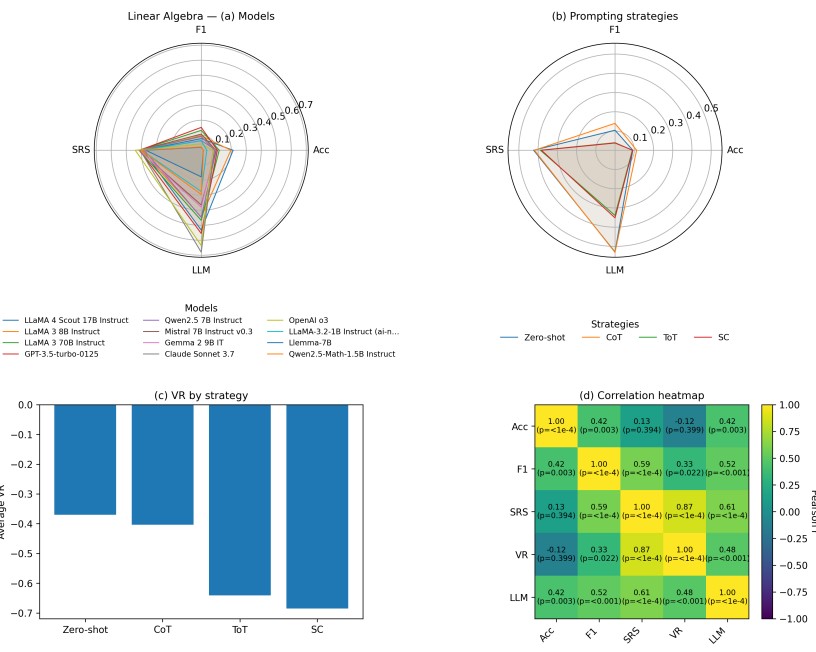

Figure 8: **Multi-metric summary of Linear Algebra results.** (a) Average scores across models. (b) Average scores across prompting methods. (c) Average VR across prompting methods. (d) Pearson correlation heatmap with $p-$values.

Table 18: **Main Results on Pre-calculus.** Evaluation of LLMs across four prompting strategies and five metrics: Accuracy (Acc), Semantic F1, SRS, VR, and LLM-based evaluation (LLM, normalized to [0,1]). The highest value in each column is **bold and underlined**.

| Model | Zero-shot | | | | | CoT | | | | | ToT | | | | | SC | | | | |
|---|---|---|---|---|---|---|---|---|---|---|---|---|---|---|---|---|---|---|---|---|
| | Acc | F1 | SRS | VR | LLM | Acc | F1 | SRS | VR | LLM | Acc | F1 | SRS | VR | LLM | Acc | F1 | SRS | VR | LLM |
| *Closed-source Models* | | | | | | | | | | | | | | | | | | | | |
| GPT-4.1 | 0.38 | 0.13 | 0.41 | -0.51 | 0.53 | 0.40 | 0.15 | 0.41 | -0.53 | 0.56 | 0.36 | 0.03 | 0.37 | -0.79 | 0.27 | 0.38 | 0.03 | 0.38 | -0.70 | 0.25 |
| GPT-3.5-turbo-0125 | 0.37 | **0.36** | 0.50 | -0.52 | 0.52 | 0.37 | **0.46** | **0.55** | **-0.32** | 0.55 | 0.33 | **0.08** | 0.40 | -0.82 | 0.57 | 0.34 | 0.08 | 0.41 | -0.83 | 0.58 |
| OpenAI o3 | 0.32 | 0.29 | **0.55** | -0.25 | 0.67 | 0.33 | 0.24 | 0.51 | -0.36 | 0.69 | 0.30 | 0.05 | **0.42** | **-0.61** | **0.60** | 0.34 | 0.07 | **0.43** | **-0.55** | 0.58 |
| Claude Sonnet 3.7 | 0.38 | 0.21 | 0.43 | -0.73 | **0.73** | 0.41 | 0.23 | 0.44 | -0.66 | **0.73** | 0.36 | 0.06 | 0.41 | -0.83 | 0.59 | 0.38 | **0.12** | 0.42 | -0.79 | **0.67** |
| *Open-source Models* | | | | | | | | | | | | | | | | | | | | |
| DeepSeek-R1-Distill-Qwen-32B | 0.39 | 0.10 | 0.41 | -0.71 | 0.63 | 0.44 | 0.10 | 0.40 | -0.70 | 0.55 | 0.39 | 0.03 | 0.38 | -0.90 | 0.31 | 0.36 | 0.02 | 0.38 | -0.87 | 0.21 |
| Gemma 2 9B IT | 0.34 | 0.19 | 0.48 | -0.43 | 0.61 | 0.37 | 0.36 | 0.52 | -0.42 | 0.63 | 0.23 | 0.07 | 0.40 | -0.77 | 0.54 | 0.08 | 0.05 | 0.42 | -0.69 | 0.40 |
| LLaMA 3 8B Instruct | 0.39 | 0.22 | 0.44 | -0.64 | 0.62 | 0.40 | 0.27 | 0.45 | -0.64 | 0.64 | 0.36 | 0.06 | 0.40 | -0.86 | 0.52 | 0.36 | 0.05 | 0.40 | -0.87 | 0.56 |
| LLaMA 3 70B Instruct | 0.38 | 0.33 | 0.54 | -0.32 | 0.41 | 0.41 | 0.40 | 0.51 | -0.44 | 0.61 | 0.38 | **0.08** | 0.39 | -0.83 | 0.57 | 0.39 | 0.08 | 0.40 | -0.82 | 0.55 |
| LLaMA 4 Scout 17B Instruct | **0.42** | 0.20 | 0.43 | -0.70 | 0.67 | **0.44** | 0.23 | 0.43 | -0.69 | 0.65 | **0.43** | **0.08** | 0.40 | -0.82 | 0.44 | **0.40** | 0.05 | 0.40 | -0.80 | 0.37 |
| Qwen2.5 7B Instruct | 0.39 | 0.15 | 0.42 | -0.63 | 0.55 | **0.44** | 0.21 | 0.42 | -0.63 | 0.56 | 0.40 | 0.04 | 0.37 | -0.83 | 0.30 | **0.40** | 0.04 | 0.38 | -0.84 | 0.38 |
| Mistral 7B Instruct v0.3 | 0.20 | 0.19 | 0.54 | **-0.16** | 0.29 | 0.30 | 0.33 | 0.50 | -0.39 | 0.40 | 0.18 | 0.06 | 0.41 | -0.81 | 0.37 | 0.24 | 0.07 | 0.40 | -0.86 | 0.40 |
| *Math-specialized Models* | | | | | | | | | | | | | | | | | | | | |
| Qwen2.5-Math-7B Instruct | 0.25 | 0.13 | 0.40 | -0.64 | 0.47 | 0.24 | 0.10 | 0.40 | -0.64 | 0.29 | 0.25 | 0.04 | 0.38 | -0.77 | 0.32 | 0.27 | 0.03 | 0.37 | -0.83 | 0.23 |
| Llemma-7B | 0.05 | 0.02 | 0.41 | -0.58 | 0.23 | 0.04 | 0.02 | 0.41 | -0.56 | 0.28 | 0.03 | 0.01 | 0.38 | -0.84 | 0.09 | 0.03 | 0.01 | 0.34 | -0.95 | 0.11 |
| Qwen2.5-Math-1.5B Instruct | 0.10 | 0.07 | 0.46 | -0.39 | 0.39 | 0.22 | 0.10 | 0.41 | -0.60 | 0.49 | 0.22 | 0.04 | 0.39 | -0.76 | 0.17 | 0.22 | 0.03 | 0.38 | -0.81 | 0.13 |
| LLaMA-3.2-1B Instruct (ft) | 0.12 | 0.20 | 0.45 | -0.59 | 0.48 | 0.11 | 0.21 | 0.45 | -0.62 | 0.33 | 0.14 | 0.06 | 0.41 | -0.83 | 0.16 | 0.12 | 0.05 | 0.41 | -0.85 | 0.13 |

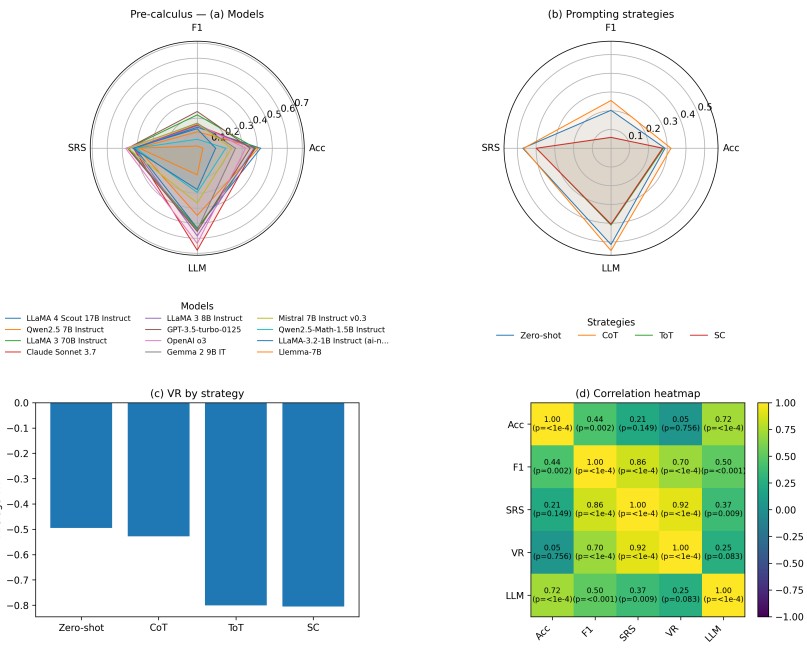

Figure 9: **Multi-metric summary of Pre-calculus results.** (a) Average scores across models. (b) Average scores across prompting methods. (c) Average VR across prompting methods. (d) Pearson correlation heatmap with $p-$values.

Table 19: **Main Results on Trigonometry.** Evaluation of LLMs across four prompting strategies and five metrics: Accuracy (Acc), Semantic F1, SRS, VR, and LLM-based evaluation (LLM, normalized to [0,1]). The highest value in each column is **bold and underlined**.

| Model | Zero-shot | | | | | CoT | | | | | ToT | | | | | SC | | | | |
|---|---|---|---|---|---|---|---|---|---|---|---|---|---|---|---|---|---|---|---|---|
| | Acc | F1 | SRS | VR | LLM | Acc | F1 | SRS | VR | LLM | Acc | F1 | SRS | VR | LLM | Acc | F1 | SRS | VR | LLM |
| *Closed-source Models* | | | | | | | | | | | | | | | | | | | | |
| GPT-4.1 | 0.35 | 0.12 | 0.39 | -0.53 | 0.72 | 0.35 | 0.13 | 0.40 | -0.53 | 0.72 | 0.34 | 0.02 | 0.37 | -0.76 | 0.46 | 0.34 | 0.02 | 0.37 | -0.73 | 0.49 |
| GPT-3.5-turbo-0125 | 0.31 | 0.44 | 0.49 | -0.28 | 0.47 | 0.31 | **0.43** | **0.49** | -0.31 | 0.45 | 0.30 | **0.09** | 0.40 | -0.77 | 0.19 | 0.30 | **0.10** | 0.41 | -0.77 | 0.21 |
| OpenAI o3 | **0.36** | 0.24 | 0.52 | -0.02 | **0.76** | **0.35** | 0.20 | 0.48 | -0.17 | 0.67 | **0.34** | 0.04 | **0.43** | -0.48 | 0.62 | **0.34** | 0.06 | **0.43** | -0.41 | 0.66 |
| Claude Sonnet 3.7 | 0.33 | 0.18 | 0.44 | -0.50 | 0.63 | 0.33 | 0.18 | 0.44 | -0.49 | 0.63 | 0.31 | 0.05 | 0.40 | -0.80 | 0.36 | 0.31 | 0.06 | 0.40 | -0.79 | 0.38 |
| *Open-source Models* | | | | | | | | | | | | | | | | | | | | |
| DeepSeek-R1-Distill-Qwen-32B | 0.26 | 0.14 | 0.44 | -0.45 | 0.55 | 0.25 | 0.10 | 0.42 | -0.61 | 0.45 | 0.24 | 0.03 | 0.40 | -0.89 | 0.20 | 0.24 | 0.01 | 0.39 | -0.92 | 0.17 |
| Gemma 2 9B IT | 0.27 | 0.19 | 0.48 | -0.24 | 0.66 | 0.28 | 0.26 | 0.47 | -0.30 | 0.63 | 0.25 | 0.06 | 0.41 | -0.65 | 0.39 | 0.24 | 0.04 | 0.41 | -0.62 | 0.38 |
| LLaMA 3 8B Instruct | 0.26 | 0.21 | 0.43 | -0.45 | 0.55 | 0.25 | 0.26 | 0.43 | -0.49 | 0.53 | 0.24 | 0.05 | 0.40 | -0.83 | 0.30 | 0.25 | 0.05 | 0.40 | -0.83 | 0.30 |
| LLaMA 3 70B Instruct | 0.29 | **0.50** | 0.54 | -0.11 | **0.76** | 0.28 | 0.38 | 0.47 | -0.33 | 0.61 | 0.27 | **0.09** | 0.38 | -0.81 | 0.33 | 0.28 | **0.10** | 0.38 | -0.80 | 0.34 |
| LLaMA 4 Scout 17B Instruct | 0.27 | 0.21 | 0.42 | -0.59 | 0.67 | 0.27 | 0.24 | 0.42 | -0.58 | 0.65 | 0.26 | 0.08 | 0.39 | -0.79 | 0.44 | 0.26 | 0.05 | 0.39 | -0.84 | 0.37 |
| Qwen2.5 7B Instruct | 0.24 | 0.19 | 0.41 | -0.57 | 0.48 | 0.24 | 0.18 | 0.42 | -0.54 | 0.50 | 0.23 | 0.03 | 0.38 | -0.80 | 0.24 | 0.23 | 0.04 | 0.38 | -0.82 | 0.23 |
| Mistral 7B Instruct v0.3 | 0.28 | 0.19 | **0.57** | **0.21** | 0.75 | 0.27 | 0.30 | 0.47 | -0.32 | 0.57 | 0.25 | 0.06 | 0.40 | -0.86 | 0.26 | 0.25 | 0.06 | 0.40 | -0.86 | 0.26 |
| *Math-specialized Models* | | | | | | | | | | | | | | | | | | | | |
| Qwen2.5-Math-7B Instruct | 0.18 | 0.18 | 0.41 | -0.55 | 0.49 | 0.17 | 0.09 | 0.41 | -0.52 | 0.43 | 0.16 | 0.05 | 0.36 | -0.80 | 0.23 | 0.15 | 0.02 | 0.35 | -0.86 | 0.18 |
| Llemma-7B | 0.12 | 0.03 | 0.40 | -0.41 | 0.39 | 0.11 | 0.02 | 0.40 | -0.29 | 0.45 | 0.10 | 0.01 | 0.38 | -0.81 | 0.16 | 0.10 | 0.01 | 0.35 | -0.93 | 0.05 |
| Qwen2.5-Math-1.5B Instruct | 0.12 | 0.11 | 0.43 | -0.38 | 0.48 | 0.10 | 0.00 | 0.40 | **0.23** | 0.74 | 0.10 | 0.00 | 0.40 | **0.23** | **0.75** | 0.10 | 0.00 | 0.40 | **0.15** | **0.70** |
| LLaMA-3.2-1B Instruct (ft) | 0.14 | 0.20 | 0.45 | -0.51 | 0.50 | 0.13 | 0.00 | 0.40 | **0.23** | **0.75** | 0.12 | 0.00 | 0.40 | **0.23** | **0.75** | 0.11 | 0.00 | 0.40 | **0.15** | **0.70** |

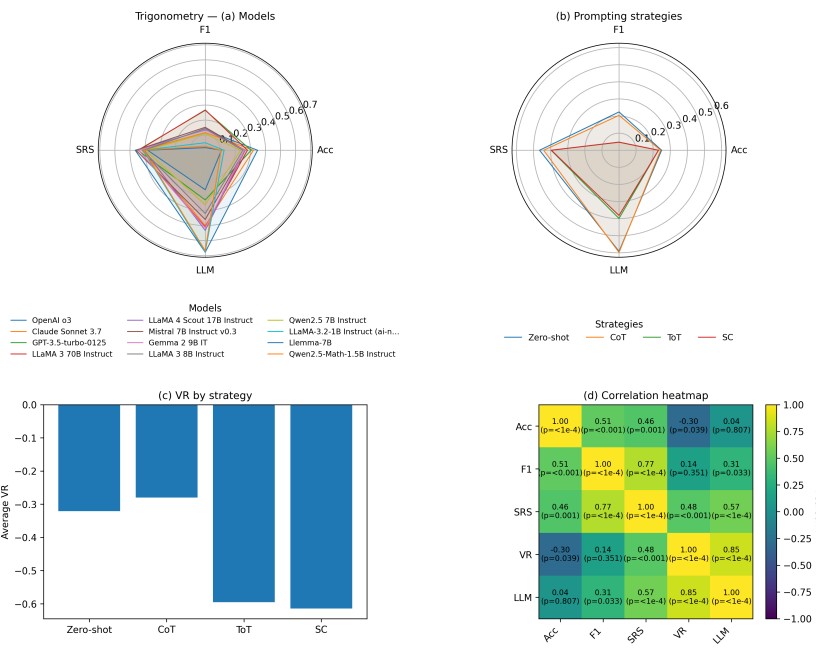

Figure 10: **Multi-metric summary of Trigonometry.** (a) Average scores across models. (b) Average scores across prompting methods. (c) Average VR across prompting methods. (d) Pearson correlation heatmap with $p-$values.

## F    INTER-RATER AGREEMENT RESULTS

Table 20: Krippendorff's $\alpha$ by topic for human–human and human–LLM ratings.

| Topic | Human–Human $\alpha$ | Human–LLM $\alpha$ |
|---|---|---|
| Calculus | 0.882 | 0.861 |
| Differential Equations | 0.602 | 0.619 |
| Discrete Mathematics | 0.680 | 0.737 |
| Linear Algebra | 0.842 | 0.881 |
| Multivariable CalculuS | 0.722 | 0.726 |
| Precalculus | 0.822 | 0.855 |
| Trigonometry | 0.926 | 0.857 |
| **Overall** | **0.829** | **0.832** |

Table 21: Pairwise quadratic Cohen's $\kappa$ for Calculus.

| Raters | Pair | $\kappa$ |
|---|---|---|
| Human–Human | H1–H2 | 0.928 |
| Human–Human | H1–H3 | 0.812 |
| Human–Human | H2–H3 | 0.860 |
| Human–LLM | H1–LLM | 0.826 |
| Human–LLM | H2–LLM | 0.886 |
| Human–LLM | H3–LLM | 0.791 |

Table 22: Pairwise quadratic Cohen's $\kappa$ for Differential Equations.

| Raters | Pair | $\kappa$ |
|---|---|---|
| Human–Human | H1–H2 | 0.624 |
| Human–Human | H1–H3 | 0.420 |
| Human–Human | H2–H3 | 0.581 |
| Human–LLM | H1–LLM | 0.647 |
| Human–LLM | H2–LLM | 0.519 |
| Human–LLM | H3–LLM | 0.573 |

Table 23: Pairwise quadratic Cohen's $\kappa$ for Discrete Mathematics.

| Raters | Pair | $\kappa$ |
|---|---|---|
| Human–Human | H1–H2 | 0.616 |
| Human–Human | H1–H3 | 0.532 |
| Human–Human | H2–H3 | 0.711 |
| Human–LLM | H1–LLM | 0.701 |
| Human–LLM | H2–LLM | 0.481 |
| Human–ILLM | H3–LLM | 0.601 |

Table 24: Pairwise quadratic Cohen's $\kappa$ for Linear Algebra.

| Raters | Pair | $\kappa$ |
|---|---|---|
| Human–Human | H1–H2 | 0.873 |
| Human–Human | H1–H3 | 0.851 |
| Human–Human | H2–H3 | 0.839 |
| Human–LLM | H1–LLM | 0.925 |
| Human–LLM | H2–LLM | 0.932 |
| Human–LLM | H3–LLM | 0.857 |

Table 25: Pairwise quadratic Cohen's $\kappa$ for Multivariable Calculus.

| Raters | Pair | $\kappa$ |
|---|---|---|
| Human–Human | H1–H2 | 0.891 |
| Human–Human | H1–H3 | 0.497 |
| Human–Human | H2–H3 | 0.786 |
| Human–LLM | H1–LLM | 0.776 |
| Human–LLM | H2–LLM | 0.866 |
| Human–LLM | H3–LLM | 0.654 |

Table 26: Pairwise quadratic Cohen's $\kappa$ for Precalculus.

| Raters | Pair | $\kappa$ |
|---|---|---|
| Human–Human | H1–H2 | 0.750 |
| Human–Human | H1–H3 | 0.657 |
| Human–Human | H2–H3 | 0.884 |
| Human–LLM | H1–LLM | 0.392 |
| Human–LLM | H2–LLM | 0.753 |
| Human–LLM | H3–LLM | 0.607 |

Table 27: Pairwise quadratic Cohen's $\kappa$ for Trigonometry.

| Raters | Pair | $\kappa$ |
|---|---|---|
| Human–Human | H1–H2 | 0.892 |
| Human–Human | H1–H3 | 0.867 |
| Human–Human | H2–H3 | 0.919 |
| Human–LLM | H1–LLM | 0.744 |
| Human–LLM | H2–LLM | 0.772 |
| Human–LLM | H3–LLM | 0.691 |

# G  LLM SOLUTIONS

In the following solution transcripts, any text highlighted in red indicates a step or claim where the model's reasoning is incorrect.

---

**Solution 1: GPT-4.1 Solution (Indefinite Integral)**

We are given the integral:
$$\int \frac{1 - \sin x}{x + \cos x} \, dx$$

---

### Step 1: Use substitution Let:
$$u = x + \cos x$$
Then:
$$\frac{du}{dx} = 1 - \sin x \quad \Rightarrow \quad du = (1 - \sin x)\, dx$$
This matches the numerator exactly, so we can rewrite the integral as:
$$\int \frac{1 - \sin x}{x + \cos x}\, dx = \int \frac{1}{u}\, du$$

### Step 2: Integrate
$$\int \frac{1}{u}\, du = \ln|u| + C = \ln|x + \cos x| + C$$

### Final Answer
$$\boxed{\int \frac{1 - \sin x}{x + \cos x}\, dx = \ln|x + \cos x| + C}$$

---

## Solution 2: GPT-4.1 Solution (Definite Integral)

We are given the definite integral:
$$I = \int_{-\pi/6}^{\pi/6} \frac{1 - \sin x}{x + \cos x}\, dx.$$

—

### Step 1: Consider symmetry
The interval is symmetric about 0: from $-\frac{\pi}{6}$ to $\frac{\pi}{6}$. Let's define the integrand:
$$f(x) = \frac{1 - \sin x}{x + \cos x}.$$
To evaluate this, let's analyze its **symmetry** by examining $f(-x)$:
$$f(-x) = \frac{1 - \sin(-x)}{-x + \cos(-x)} = \frac{1 + \sin x}{-x + \cos x}.$$
So,
$$f(-x) = \frac{1 + \sin x}{-x + \cos x}.$$
This doesn't look like an even or odd function directly. But we **add** $f(x)$ and $f(-x)$ to explore further:
—

### Step 2: Add $f(x) + f(-x)$
$$f(x) + f(-x) = \frac{1 - \sin x}{x + \cos x} + \frac{1 + \sin x}{-x + \cos x}.$$
Let's write it as a single expression:
Let's denote:
$$A = \frac{1 - \sin x}{x + \cos x}, \quad B = \frac{1 + \sin x}{-x + \cos x}.$$
Common denominator is:
$$(x + \cos x)(-x + \cos x) = \cos^2 x - x^2.$$
So we get:

$$A + B = \frac{(1 - \sin x)(-x + \cos x) + (1 + \sin x)(x + \cos x)}{\cos^2 x - x^2}.$$

Now expand numerator:

- $(1 - \sin x)(-x + \cos x) = -x(1 - \sin x) + \cos x(1 - \sin x) = -x + x \sin x + \cos x - \cos x \sin x$
- $(1 + \sin x)(x + \cos x) = x(1 + \sin x) + \cos x(1 + \sin x) = x + x \sin x + \cos x + \cos x \sin x$

Add both:

$$\text{Numerator} = (-x + x \sin x + \cos x - \cos x \sin x) + (x + x \sin x + \cos x + \cos x \sin x)$$

Simplify:

- $-x + x = 0$
- $x \sin x + x \sin x = 2x \sin x$
- $\cos x + \cos x = 2 \cos x$
- $-\cos x \sin x + \cos x \sin x = 0$

So total numerator is:

$$2x \sin x + 2 \cos x = 2(x \sin x + \cos x).$$

Therefore,

$$f(x) + f(-x) = \frac{2(x \sin x + \cos x)}{\cos^2 x - x^2}.$$

—

### Step 3: Now integrate $f(x) + f(-x)$ over $-\pi/6$ to $\pi/6$

From integral properties over symmetric intervals:

$$\int_{-a}^{a} f(x)\, dx = \int_{-a}^{a} \frac{1}{2}[f(x) + f(-x)]\, dx.$$

So:

$$I = \int_{-\pi/6}^{\pi/6} f(x)\, dx = \frac{1}{2} \int_{-\pi/6}^{\pi/6} f(x) + f(-x)\, dx = \frac{1}{2} \int_{-\pi/6}^{\pi/6} \frac{2(x \sin x + \cos x)}{\cos^2 x - x^2}\, dx$$

Simplify:

$$I = \int_{-\pi/6}^{\pi/6} \frac{x \sin x + \cos x}{\cos^2 x - x^2}\, dx.$$

—

### Step 4: Check whether the integrand is **odd**

Let:

$$g(x) = \frac{x \sin x + \cos x}{\cos^2 x - x^2}.$$

Compute $g(-x)$:

- Numerator: $-x \sin x + \cos x$
- Denominator: $\cos^2 x - x^2$ (even function)

So,

$$g(-x) = \frac{-x \sin x + \cos x}{\cos^2 x - x^2} = -\frac{x \sin x + \cos x}{\cos^2 x - x^2} = -g(x)$$

Thus, $g(x)$ is an odd function.

—

### Step 5: Integral of an odd function over symmetric interval

Since $g(x)$ is odd and the limits are symmetric:

$$\int_{-a}^{a} g(x)\,dx = 0$$

So,

$$I = \int_{-\pi/6}^{\pi/6} f(x)\,dx = \int_{-\pi/6}^{\pi/6} g(x)\,dx = 0$$

—

### Final Answer:

$$\boxed{0}$$

---

**Solution 3: OpenAI-o3 Solution (Definite Integral)**

We are asked to evaluate the definite integral

$$\int_{-\pi/6}^{\pi/6} \frac{1 - \sin(x)}{x + \cos(x)}\,dx.$$

First, observe that this integrand does not have a standard elementary antiderivative due to the complexity of the expression and the interaction between the trigonometric and algebraic terms. Therefore, symbolic integration is not practical.
Next, we examine the symmetry of the integrand. Define

$$f(x) = \frac{1 - \sin(x)}{x + \cos(x)}.$$

Evaluating $f(-x)$ gives:

$$f(-x) = \frac{1 + \sin(x)}{-x + \cos(x)},$$

which is not equal to $f(x)$ or $-f(x)$, so the function is neither even nor odd. Thus, no symmetry can be used to simplify the computation.
Since an analytical solution is unavailable, we use a numerical method. Specifically, we apply the Trapezoidal Rule with $n = 6$ subintervals over the interval $[-\pi/6, \pi/6]$. The width of each subinterval is

$$h = \frac{\pi/3}{6} = \frac{\pi}{18}.$$

Using the Trapezoidal Rule:

$$\int_{a}^{b} f(x)\,dx \approx \frac{h}{2}\left[f(x_0) + 2f(x_1) + 2f(x_2) + \cdots + 2f(x_{n-1}) + f(x_n)\right].$$

After performing the calculation, we find the approximate value of the integral to be:

$$\int_{-\pi/6}^{\pi/6} \frac{1 - \sin(x)}{x + \cos(x)}\,dx \approx 1.4511.$$

Therefore, the final answer is approximately

$$\boxed{1.4511}.$$

> **Solution 4: Mistral 7B Instruct v0.3 Solution (Definite Integral)**
>
> To solve this integral, we can't use a simple substitution or integration by parts, so we'll use numerical methods to approximate the value. We'll use the trapezoidal rule, which is a numerical integration method that provides a good approximation for definite integrals. The trapezoidal rule formula is:
>
> $$T = h * [f(a) + 2 * \Sigma(f(x_i) \text{ for i = 1 to n}) + f(b)]/n$$
>
> where $h = (b - a)/n$, $n$ is the number of subintervals, and $f(x)$ is the function we're integrating.
> Let's choose $n = 1000$ for a good approximation.
> First, let's find h:
>
> $$h = (\pi/6 - (-\pi/6))/1000 = \pi/(3000 * 2) = \pi/6000$$
>
> Now, let's find the subintervals' endpoints:
>
> $$x_i = -\pi/6 + i * h \quad \text{for i = 0 to 1000}$$
>
> Next, we'll calculate the function values for each subinterval:
>
> $$f(x_i) = (1 - \sin(x_i))/(x_i + \cos(x_i)) \quad \text{for i = 0 to 1000}$$
>
> Finally, we'll sum up the function values and add the endpoints' values:
>
> $$T = h * [f(-\pi/6) + 2 * \Sigma(f(x_i) \text{ for i = 1 to 1000}) + f(\pi/6)]$$
>
> After calculating the sum and multiplying by h, we get the approximate value of the integral:
>
> $$\boxed{T \approx 1.400731}$$
>
> This is the approximate value of the definite integral $\int_{-\pi/6}^{\pi/6} \frac{1-\sin x}{x+\cos x}\, dx$ using the trapezoidal rule with n = 1000.

## H  EXAMPLE OF EVALUATIONS

### H.1  EVALUATION OF A CORRECT SOLUTION

This evaluation corresponds to the GPT-4.1 solution of the *indefinite integral* presented in Appendix 1.

Table 28: Automatic metric evaluation for GPT-4.1 solution (zero-shot) on an *indefinite integral*.

| Model | Prompt | Accuracy | F1 | SRS | VR |
|---|---|---|---|---|---|
| GPT-4.1 | Zero-shot | 1 | 0.8 | 0.661 | 0.396 |

Table 29: Breakdown of Stepwise Reasoning Score (SRS) for GPT-4.1 solution (zero-shot) on the *indefinite integral*. Higher values indicate better logical structure, informativeness, and consistency.

| Breakdown SRS | Score |
|---|---|
| Faithfulness | 0.742 |
| Informativeness (Step) | 0.698 |
| Informativeness (Chain) | 0.939 |
| Coherence (Step vs. Step) | 0.096 |
| Discourse Representation | 0.726 |
| Repetition (Step) | 0.762 |
| **Average (SRS)** | **0.661** |

Table 30: Automatic grader evaluation for GPT-4.1 solution (zero-shot) on the *indefinite integral* solution.

| Step | Description | Auto-Grader Evaluation | Score (/5) |
|---|---|---|---|
| 1 | Restates the integral $\int \frac{1-\sin x}{x+\cos x}\, dx$ | Correct setup; clear starting point, but does not yet advance the solution. | 4/5 |
| 2 | Substitution $u = x + \cos x$, with $du = (1 - \sin x)\, dx$ and rewrite to $\int \frac{du}{u}$ | Substitution and differential are correct; rewriting to $u$-form is accurate and clearly explained. | 5/5 |
| 3 | Integrate and back-substitute: $\int \frac{du}{u} = \ln|u| + C = \ln|x + \cos x| + C$ | Antiderivative and back-substitution are correct; final boxed answer is clearly presented. | 5/5 |
| | | **Final Score** | **4.67 / 5** |

## H.2 EVALUATION OF AN INCORRECT SOLUTION

This evaluation corresponds to the GPT-4.1 solution of the *definite integral* presented in Appendix 2.

Table 31: Automatic metric evaluation for GPT-4.1 Solution (Zero-shot) on an *definite integral*.

| Model | Prompt | Accuracy | F1 | SRS | VR |
|---|---|---|---|---|---|
| GPT-4.1 | Zero-shot | 0 | 0.44 | 0.509 | 0.264 |

Table 32: Breakdown of Stepwise Reasoning Score (SRS) for GPT-4.1 Solution (Zero-shot) on the *definite integral*. Higher values indicate better logical structure, informativeness, and consistency.

| Breakdown SRS | Score |
|---|---|
| Faithfulness | 0.715 |
| Informativeness (Step) | 0.694 |
| Informativeness (Chain) | 0.913 |
| Repetition (Step) | 0.065 |
| Discourse Representation | 0.622 |
| Coherence (Step vs. Step) | 0.045 |
| **Average** | **0.638** |

Table 33: Automatic grader evaluation of the GPT-4.1 Solution (Zero-shot) on the **definite** integral.

| Step | Description | Auto-Grader Evaluation | Score (/5) |
|------|-------------|------------------------|------------|
| 1 | Restates $I = \int_{-\pi/6}^{\pi/6} \frac{1-\sin x}{x+\cos x}\, dx$ | Clear restatement; good foundation for the solution. | 5/5 |
| 2 | Notes symmetry of limits; computes $f(-x)$ for $f(x) = \frac{1-\sin x}{x+\cos x}$ | Symmetry identified; $f(-x)$ computed correctly. | 5/5 |
| 3 | Forms $f(x) + f(-x)$ and simplifies to $\frac{2(x\sin x + \cos x)}{\cos^2 x - x^2}$ | Algebra and simplification are valid and carefully done. | 5/5 |
| 4 | Uses $\int_{-a}^{a} f(x)\, dx = \frac{1}{2}\int_{-a}^{a}(f(x) + f(-x))\, dx$ to get $\int \frac{x\sin x + \cos x}{\cos^2 x - x^2}\, dx$ | Proper use of symmetry to rewrite the integral. | 5/5 |
| 5 | Claims $g(x) = \frac{x\sin x + \cos x}{\cos^2 x - x^2}$ is odd | Incorrect: numerator at $-x$ is $-x\sin x + \cos x$, not the negative of the original; denominator is even $\Rightarrow g$ is *not* odd. | 1/5 |
| 6 | Concludes $I = 0$ from "odd integrand over symmetric limits" | Conclusion depends on the incorrect oddness claim, so the result is wrong. | 1/5 |
| | | **Final Score** | **3.67 / 5** |

