# OpenReview forum: "CUMath: A Benchmark and Evaluation Framework for LLMs on Mathematical Reasoning in Undergraduate Computational Math"
_ICLR.cc/2026/Conference — Submitted to ICLR 2026_

### Official Review · Reviewer_Ckz3 · 2025-10-23

**Soundness:** 2
**Presentation:** 3
**Contribution:** 2
**Rating:** 6
**Confidence:** 3

**Summary:**

This paper introduces a dataset called CUMath which can be used as a benchmark of 2,100 real problems from undergraduate courses in Calculus, Differential Equations, Discrete Mathematics, Linear Algebra, Multivariable Calculus, Precalculus, and Trigonometry. Each problem includes step-by-step solutions, enabling evaluation of both final answers and intermediate reasoning. It categorize the problems into three answer formats: Free Response (FR), Short Answer (SA), and True/False (TF).

The motivation behind releasing this dataset stems from the observation that current generation LLMs perfom well on existing popular math benchmark dataset but still they strggle with math reasoning.

Further, this paper proposes a multi-layered framework to jointly evaluate answer accuracy and the reasoning of the model.  This framework combines automatic metrics with an LLM-as-a-grader pipeline.

Through the propsed CUMath dataset and evaluation framework, this paper shows that SOTA LLMs make mistakes in the symbolic manipulation and procedural reasoning even when prducing final correct answer.

**Strengths:**

- The dataset is balanced across sever core subjects of the Maths so that no single subject dominates or remains underrepresented.
- This paper proposes a multi-layered evaluation framework where it combines MathBERT for symbolic encoding, LLM for step-level reasoning assessment, and Wolfram Alpha for answer verification. This pipelines, thus, captures both answer correctness and reasoning quality.
- The proposed benchmarking dataset in this paper would be valuable towards advancing the SOTA of LLM’s reasoning abilities.
- Sections 6.1 and 6.2 provide interesting insights about LLMs behavior at large when they fail in math reasoning.

**Weaknesses:**

- The technical novelty of the paper is limited but that is understandable because it is more of a data set contribution paper.
- It will be good to see the quantitative comparison of the proposed Dataset+Eval framework against well known Math reasoning dataset under the same proposed eval framework and the same set of models. This will help readers buy the key selling points of the paper. See my comment in the Questions section also.

**Questions:**

- In Section 4.1, why use two different notations $\hat{s}_i^j$ and $e_i^j$ for the same thing.
- In Line 228, the quantity $m_k(e_i)$ is not defined.
- In Line 269, it is written that “The encoded steps are passed to an LLM..” I was wondering how do you pass embeddings to an LLM? Can you elaborate? My understanding is that you are passing embeddings obtained from MathBERT (in Step 2) to an LLM in Step 3.
- In Table 2, it will be good if you can also add the performance of these models for some of the popular Math reasoning datasets but using your eval framework. This will help readers buy the points that your trying to drive home.

---

> ### Author Response · Authors · 2025-11-17
>
> Thank you for the detailed questions and constructive suggestions. We address each point below.
>
> **1. Notation in Section 4.1**:
>
> The two notations are intentionally distinct because they serve different roles in our evaluation pipeline as follows.
>  - We denote the generated reasoning steps as the unordered set $\hat{S}_i = \{\hat{s}_i^{1}, \ldots, \hat{s}_i^{\hat{n}_i}\}$, which is the form required for the Semantic F1 computation. Semantic F1 performs pairwise matching over steps using cosine similarity, and this matching is set-based, that is, it does not depend on the original order of the steps, only on the existence of semantic matches. Representing the steps as a set $\hat{S}_i$ makes this explicit.
> - For all other reasoning-quality metrics in Section 4, the ordering of steps is essential. For that purpose, we additionally introduce the ordered sequence $e_i = (e_i^{1}, e_i^{2}, \ldots, e_i^{n_i}) = (\hat{s}_i^{1}, \hat{s}_i^{2}, \ldots,  \hat{s}_i^{\hat{n}_i})$, which is the same underlying collection of steps, but now viewed explicitly as an ordered sequence.
>
> To summarize,
> - $\hat{S}_i$: unordered set of model-generated steps (used for Semantic F1 matching)
> - $e_i$: ordered sequence of the same steps (used for stepwise reasoning metrics).
>
> **2. Undefined quantity in Line 228**
>
> The quantity was indeed missing a formal definition. We have now added the correct definition to our manuscript.
>
> **3. How embeddings are passed to the LLM (Line 269)**
>
> We do not feed MathBERT embeddings directly as input tokens to the LLM. Instead, the embeddings are used as conditioning features within the evaluation prompt. We convert the embeddings into a textual representation that is included in the prompt. This allows the LLM to evaluate based on the given condition of semantic information, and it encodes without requiring any architectural changes. We have revised the manuscript to clarify this workflow and to avoid the impression that embeddings are injected into the LLM’s internal layers.
>
> **4. Suggestion for Table 2**
>
> Adding results on standard math-reasoning datasets using our evaluation framework would indeed help support our claims. However, our current work focuses on presenting the evaluation method itself, and many existing math datasets do not include the step-by-step reasoning needed for our metrics, so we are not able to provide these direct comparisons at this time.
>
> That said, this is an important direction for future work. Our paper already shows that even models with the same accuracy can differ significantly in the quality and correctness of their intermediate steps. Therefore, we believe that evaluating mathematical reasoning should extend beyond final-answer accuracy, and future researchers should incorporate step-level evaluation frameworks like ours to more accurately assess the quality of models' output.

---

> ### Comment · Reviewer_Ckz3 · 2025-11-22
> **Response to Authors' Comments**
>
> Thanks for taking time to respond. Things are quite clear now.

---

### Official Review · Reviewer_ELDK · 2025-10-31

**Soundness:** 2
**Presentation:** 2
**Contribution:** 2
**Rating:** 4
**Confidence:** 4

**Summary:**

The paper aims to address two core issues in evaluating the mathematical reasoning capabilities of Large Language Models (LLMs) at the undergraduate level: (1) existing benchmarks are either too elementary or too advanced, lacking diagnostic value for reasoning failures, and (2) current evaluation paradigms often decouple final answer accuracy from the quality of the reasoning process. To this end, the authors introduce CUMath, a new benchmark of 2,100 problems from real undergraduate computational math courses, with each problem annotated with step-by-step solutions. Concurrently, they propose a multi-layered evaluation framework that combines automatic metrics with an "LLM-as-a-grader" pipeline, which is augmented with external tools like MathBERT and Wolfram Alpha for verification. Through an evaluation of 15 LLMs, the authors conclude that even frontier models exhibit systematic errors in symbolic manipulation and procedural reasoning, arguing for the necessity of integrated evaluations that assess both reasoning validity and answer correctness.

**Strengths:**

The paper accurately identifies a critical gap in the current landscape of LLM evaluation. As models approach saturation on benchmarks like GSM8K, there is a pressing need for more challenging, realistic, and diagnostically useful benchmarks. The focus on undergraduate mathematics is an excellent choice for a domain that can effectively distinguish between superficial fluency and deep reasoning abilities.
The construction of the CUMath dataset is a solid and meaningful effort. Its grounding in authentic instructional materials (quizzes, exams, textbooks) ensures the practical relevance of the problems. Most importantly, providing detailed step-by-step solutions is crucial for enabling fine-grained analysis of where models' reasoning chains fail, which will greatly benefit future research in this area.

The authors' advocacy for an integrated assessment of "answer correctness" and "reasoning quality" is insightful. Highlighting the evaluation blind spot of "correct answers derived from flawed reasoning" demonstrates a deep understanding of the limitations of current LLM evaluation. The conceptual direction of the proposed multi-layered framework, which attempts to synthesize automated metrics with qualitative LLM-based feedback, is both correct and worthy of exploration.

**Weaknesses:**

The authors repeatedly claim to evaluate "state-of-the-art LLMs" or "frontier LLMs." However, the list of evaluated models (Table 12) primarily consists of older models such as GPT-3.5, an early version of GPT-4.1, and smaller-scale open-source models (e.g., LLaMA 3 8B/70B). Given the rapid pace of development in the field (and a target publication date of ICLR 2026), these models are no longer representative of the cutting edge. More recent and powerful reasoning models, such as the latest GPT and Claude series or other specialized math models, are conspicuously absent. This outdated selection invalidates the paper's main conclusion that "even the strongest LLMs achieve an accuracy of less than 25%." A rigorous claim about the capabilities of "frontier models" must be substantiated by testing the models widely considered to be the most capable at the time of submission. Without such experiments, the observed failures could be limitations of the specific models tested rather than a general bottleneck for all LLMs.
While the concept of a "multi-layered evaluation framework" is appealing, its components are largely direct applications or combinations of existing work (e.g., SRS from ROSCOE, VR from ReasonEval). The main claimed novelty, the "LLM-as-a-grader" pipeline, lacks the most critical piece of validation: there is no quantitative analysis comparing its outputs to those of human experts. A reliable automated grading system must demonstrate high inter-rater reliability (e.g., using Cohen's Kappa or Krippendorff's Alpha) with human graders. Without this evidence, the reliability and fairness of the LLM grader cannot be trusted, rendering the scores it produces (the "LLM" column in Table 2) unsubstantiated.

The framework's reliability is highly dependent on its automated preprocessing modules, especially the "Math Segmentation" component. As described in Section 4.2, this module relies on simple heuristics—looking for explicit "step k" markers and defaulting to "line-based segmentation" otherwise. This approach is extremely brittle when processing the free-form, structurally diverse outputs of LLMs. A single complex reasoning step can span multiple lines, and models may use different delimiters or none at all. Incorrect segmentation leads to cascading errors in all subsequent evaluation steps (e.g., semantic F1, SRS, LLM-as-a-grader). Yet, the paper provides no evaluation of this module's accuracy (e.g., against a human-annotated ground truth) nor does it discuss its fault tolerance or potential impact on the final results. This oversight regarding a core component's robustness casts serious doubt on the entire framework's practical usability.

**Questions:**

1.	Could you explain the decision to exclude more recent, top-performing models renowned for their mathematical reasoning abilities (e.g., the latest GPT-4 series, Claude 3 series)? Given that your central conclusion is about the upper-bound capabilities of "frontier LLMs," how can this claim be supported by the current selection of models?
2.	Do you have any plans to conduct, or have you already performed, a study comparing the ratings from your "LLM-as-a-grader" pipeline against scores from human mathematics experts? Without such a comparison, how do you ensure the reliability and impartiality of the automated grader, preventing it from being merely a black box?
3.	Have you evaluated the accuracy of the "Math Segmentation" module? What is its error rate on LLM outputs that lack explicit step markers or follow non-standard formatting? How significantly do these potential segmentation errors impact the downstream F1, SRS, and LLM-grader scores?

---

> ### Author Response · Authors · 2025-11-21
>
> Thank you for your detailed questions and constructive suggestions. We address each point below.
>
> **1. Choice of evaluated models and the “frontier LLMs” claim**
>
> Our experiments were conducted during a fixed evaluation window (May-August 2025). At the time, GPT-4.1 (released in April 2025) and Claude 3.7 (released in February 2025) were the most recent models in their respective series available at the time, and we selected them based on this. We have now updated the paper to clarify that when we say “frontier models,” we mean the state-of-the-art models available at the time we conducted our experience, not every newly released model at the time of public release or conference submission.
>
> We do not believe our conclusions about model mistakes depend on any single model version. The mistakes we highlight appear consistently across models, so these errors are not model-specific. Because new LLMs are released very quickly, it is impractical to rerun the entire benchmark every time a new model is introduced. It would also be challenging for us to rerun large-scale experiments during the rebuttal period due to time constraints and resource limitations. However, if the paper is accepted, we can evaluate the latest models available at that time and include those results in the camera-ready version, so that readers see the most up-to-date numbers.
>
> We also note that even if we selected the best models today, a few months later, there will almost certainly be more advanced models. This is exactly why we built CUMath: to provide a stable benchmark that the community can reuse to test new models as they are released, without requiring us to redo the entire study each time. Therefore, we would like to ask the reviewer whether this explanation of our model selection and scope addresses the concern, or whether they feel additional large-scale experiments are strictly necessary.
>
> **2. Validation of the LLM-as-a-Grader through human graders' comparison**
>
> In the revised manuscript, we have conducted exactly this comparison. As described in Section 4.3 and Section 5.3, and detailed in Appendix F, we performed a human evaluation study that directly compares the LLM-as-a-grader’s scores with those from human annotators. In total, three independent human graders evaluated the same 105 solutions sampled uniformly across topics and model families using the same rubric as the automated pipeline.
>
> Our results show that the LLM-as-a-grader aligns closely with human graders: human-human agreement reached a Krippendorff’s $\alpha = 0.829$, while human-LLM agreement was nearly identical at $\alpha = 0.832$. Topic-level $\alpha$ and weighted Cohen’s $\kappa$ values show the same pattern. These findings provide empirical evidence that the LLM-as-a-grader aligns closely with human graders in evaluating solution quality.
>
> Additionally, to prevent the LLM-as-a-grader from behaving as a black box and to minimize errors or hallucinations from relying only on the LLM’s judgment, the pipeline integrates symbolic encoding (MathBERT) and external CAS verification. When the LLM’s assessment disagrees with the CAS, a revision step is enforced so that the final score reflects correct mathematical reasoning rather than model-generated errors.
>
> Overall, the human evaluation and the incorporation of symbolic encoding and computational verification steps provide clear support for the reliability and fairness of our LLM-as-a-grader pipeline. Although further evaluations can be explored in future work, our current results indicate that our grading pipeline aligns closely with human graders in evaluating solutions in computational undergraduate mathematics.
>
> **3. Math Segmentation Module**
>
> We agree that segmentation quality is important because it affects the step-based metrics. In our study, we incorporate and adapt the segmentation practices from established frameworks like ROSCOE and ReasonEval so that our evaluation remains aligned with prior work. The LLM-as-a-grader is also designed not to penalize minor formatting variations or small shifts in step boundaries, which helps limit the impact of imperfect segmentation.
>
> That said, we acknowledge that this simple segmentation approach is a limitation of our work. We did not evaluate how well our method handles outputs with unusual or inconsistent formatting, and more semantic or model-based segmentation methods could likely provide more reliable step boundaries. Exploring these stronger approaches is a natural direction for future work. Segmentation itself is an active research problem, and we do not aim to solve that problem in this paper. However, we see them as complementary to our benchmark. For example, recent work, such as “R1-Compress: Long Chain-of-Thought Compression via Chunk Compression and Search” and "StepWiser: Stepwise Generative Judges for Wiser Reasoning", proposes stronger segmentation strategies that could be integrated into future versions of our framework.

---

### Official Review · Reviewer_ApXz · 2025-11-02

**Soundness:** 2
**Presentation:** 3
**Contribution:** 2
**Rating:** 4
**Confidence:** 4

**Summary:**

This paper introduces CUMath, a benchmark to evaluate LLM reasoning in undergraduate computational math. The authors state a diagnostic gap exists, as LLMs struggle with fundamental undergraduate tasks and existing datasets are either trivial, synthetic, or overly advanced. To address this, the authors provide a dataset of 2,100 problems, each with step-by-step solutions for evaluation. The paper also proposes a multi-layered evaluation framework that integrates automatic metrics with an LLM-as-a-grader pipeline. This pipeline uses MathBERT for symbolic encoding and external verification with Wolfram Alpha. The authors' analysis of 15 LLMs shows models misuse symbolic methods and rely on shortcuts, leading to polished but flawed solutions. The findings reveal failure modes, including invalid reasoning leading to correct results, and show accuracy alone is an insufficient measure of mathematical competence.

**Strengths:**

1.  The paper clearly identifies a diagnostic gap in current math benchmarks (being either trivial or overly advanced) for evaluating LLMs.
2.  The inclusion of detailed step-by-step solutions enables fine-grained evaluation of model reasoning processes.
3. The proposed LLM-as-a-grader framework offers a novel evaluation perspective by using external CAS verification loops.

**Weaknesses:**

1.  The low accuracy (less than 25% for even the best models) makes meaningful performance comparisons between models difficult, and we still do not have a comprehensive metrics to evaluate the ability of each model.
2. The paper shows a significant divergence between automatic metrics (like Accuracy) and its own LLM-as-a-grader score , and argues the LLM score is more comprehensive. However, this claim is weakened because the paper does not report consistency data between its LLM-as-a-grader framework and human expert scores.

**Questions:**

1.  What is the agreement (e.g., kappa score) between your LLM-as-a-grader and human experts on a subset of CUMath?
2. Given the divergence between automatic metrics (like Accuracy) and LLM-as-a-grader scores, why should the LLM-grader be trusted as a comprehensive measure without a reported correlation to human expert evaluation?
3. The results table  shows that smaller open-source models (e.g., LLaMA 4 Scout 17B Instruct) achieve scores similar to top-tier models (e.g., OpenAI o3). Does this lack of differentiation suggest the benchmark is not reliably capturing capability differences, or is this an intended finding?

---

> ### Author Response · Authors · 2025-11-21
>
> Thank you for your detailed questions and constructive suggestions. We address each point below.
>
> **1. Agreement between the LLM-as-a-arader and Human Graders**
>
> We have added a human evaluation study in the revised manuscript. The methodology is described in Section 4.3, and the main quantitative findings are summarized in Section 5.3. The detailed results, including full agreement tables and topic-level breakdowns, are provided in Appendix F. In total, three independent graders with sufficient mathematical background evaluated 105 model-generated solutions uniformly sampled across topics and model families. Each solution was independently evaluated using the same rubric and criteria as the LLM-as-a-grader.
>
> Our results show strong alignment between the LLM-as-a-grader and human graders. Human-human agreement reached a Krippendorff’s $\alpha = 0.829$, while human-LLM agreement was nearly identical at $\alpha = 0.832$. Topic-level~$\alpha$ values ranged from $0.60$-$0.93$ for human-human and $0.62$-$0.88$ for human-LLM comparisons, with weighted Cohen’s $\kappa$ following the same pattern. These results indicate that the LLM-as-a-grader aligns well with human judgments on our evaluation subset and preserves the overall reliability of the scoring process.
>
> **2. Why the LLM-as-a-Grader can be trusted as a comprehensive metric**
>
> Because accuracy alone fails to capture reasoning quality, we introduced the LLM-as-a-grader to bridge this gap, and our human study now provides empirical validation of its reliability. As shown in our revised manuscript, the LLM-as-a-grader aligns closely with human evaluators on our evaluation subset. This provides evidence that it captures aspects of reasoning quality that human graders also identify.
>
> Thus, Figure 2(d) in our paper shows that the LLM-as-a-grader maintains moderate correlations with both final-answer and reasoning-based metrics, suggesting that it integrates information from different aspects. This makes it more closely aligned with how humans evaluate completed solutions, especially in cases where other metrics may over-penalize stylistic differences or under-capture flawed reasoning that leads to correct answers.
>
> **3. Similar scores for small open-source models vs. top-tier models**
>
> It is true that accuracy scores are very close across models, with smaller open-source models sometimes performing almost the same as top-tier closed-source ones. However, this highlights the limitations of accuracy alone, rather than a weakness of the dataset.
>
> Accuracy only tells us whether the final answer is correct. It does not tell us how the model got there or whether the reasoning is consistent. This motivated us to combine multiple metrics in our evaluation framework to evaluate those models. When we look at these process-based metrics, the differences between models become clearer than what accuracy alone would suggest. The scores on these metrics vary considerably across models, and this shows that there is a difference in the models' reasoning quality and completeness than what accuracy alone shows.
>
> The differences between models also become apparent when we conduct a human evaluation on a subset of the data. All human graders observed clear differences in the quality and structure of the solutions produced by different model families. It was also surprising and interesting for us that some smaller open-source models generated very strong mathematical reasoning, and even performed better than certain closed-source models on some of the problems in this subset. However, their main limitation is that they tend to use more tokens compared to closed-source models. These observations of us are also consistent with findings from recent work, such as "OckBench: Measuring the Efficiency of LLM Reasoning" (Du et al., 2025) and "Measuring Thinking Efficiency in Reasoning Models: The Missing Benchmark" (Tim, 2025). Therefore, this can make open-source models slower and less efficient to use in practice, especially for more complex problems that require longer reasoning.

---

### Author Response · Authors · 2025-11-21
**Overall Response to Reviewers**

We are thankful to all reviewers for carefully reading our submission and for their valuable feedback and suggestions. The comments raised three key issues: (1) the need for human validation of our LLM-as-a-grader framework, (2) clarification of how we chose and evaluated the models, and (3) concerns about the reliability of components such as segmentation and reasoning metrics. We have addressed each of these points in the revised manuscript.

**1. Human validation of the LLM-as-a-grader.**

A central concern from multiple reviewers was whether the automated grader could be trusted without comparison to human graders. In the revised version, we added human evaluation components (Sections 4.3 and 5.3, Appendix F). We recruited three independent annotators with sufficient mathematical background to evaluate multi-step reasoning at the undergraduate level. All annotators were instructed to use the same scoring rubric as our automatic grading pipeline. In total, they manually graded 105 model-generated solutions uniformly sampled across topics. Results show strong agreement: human–human Krippendorff’s $\alpha = 0.829$ and human–LLM $\alpha = 0.832$, with topic-level $\alpha$ and Cohen’s $\kappa$ showing consistent patterns. This demonstrates that the LLM-as-a-grader aligns closely with human graders and preserves inter-rater reliability.

To avoid hallucinations or errors from relying only on the LLM, our pipeline integrates symbolic encoding (MathBERT) and external CAS verification (Wolfram Alpha), with enforced revision when conflicts arise. The study of inter-rater consistency and the verification steps help increase confidence in the transparency and reliability of our pipeline, though we acknowledge this as an ongoing area for refinement.

**2. Clarification of evaluated models and “frontier LLMs.”**

One reviewer asked why recently released models were not included. Our experiments were conducted within a fixed evaluation window (May-August 2025), during which GPT-4.1 and Claude~3.7 were state-of-the-art models available at the time. We have clarified this explicitly in our revised manuscript. It is also important to note that the failure modes we identify, such as incorrect symbolic manipulation, surface-level reasoning patterns, and inconsistent reasoning, appear across all model families and are not specific to a single model. If accepted, we will evaluate the newest models available at camera-ready time.

**3. Robustness of segmentation and reasoning metrics.**

One reviewer noted the importance of segmentation accuracy and its impact on step-level metrics. We clarify that our segmentation procedure follows established practices from ROSCOE and ReasonEval. Furthermore, our LLM-as-a-grader design intentionally avoids over-penalizing minor formatting or boundary differences. Nonetheless, we acknowledge this module as a limitation and discuss how more advanced semantic segmentation methods (e.g., R1-Compress (Wang et al., 2025) and StepWiser (Xiong et al., 2025)) could be incorporated in future work.

**4. Clarifying metric interpretability and model comparisons.**

A reviewer pointed out that small open models and top-tier models achieved similar accuracy levels. We highlight that this is precisely why final-answer accuracy is insufficient: it compresses performance and hides differences in reasoning quality. In contrast, our process-based metrics (Semantic F1, SRS, VR) and human-aligned LLM-as-a-grader reveal substantially clearer differences. Our study also shows that human graders consistently perceived differences in solution quality across models, even when accuracy was similar.

**References:**

[1] Wang, Y., Luo, H., Yao, H., Huang, T., He, H., Liu, R., Tan, N., Huang, J., Cao, X., Tao, D., & Shen, L. (2025). R1-Compress: Long Chain-of-Thought Compression via Chunk Compression and Search.

[2] Xiong, W., Zhao, W., Yuan, W., Golovneva, O., Zhang, T., Weston, J., & Sukhbaatar, S. (2025). StepWiser: Stepwise Generative Judges for Wiser Reasoning.

---

### Meta-Review · Area_Chair_vxot · 2026-01-07

**Summary:**

This paper introduces a benchmark of 2100 undergraduate-level computational mathematics problems, annotated step by step, to address the diagnostic gap between elementary and overly advanced math benchmarks. The proposed multi-layered evaluation that combines standard automatic metrics with an LLM-as-a-grader pipeline, augmented by symbolic encoding and external verification. Experiments show that even strong models frequently exhibit flawed symbolic and procedural reasoning. The reviewers' main concerns (which this AC considers particularly significant) can be summarized as follows:

- Using outdated LLMs (Reviewer ELDK)
- Missing comparisons with human experts, providing insufficient justification for trusting the LLM-as-a-grader as a comprehensive evaluation measure in the absence of human expert results (Reviewers ApXz, ELDK)
- Existing metrics used for the framework (e.g., SRS from ROSCOE and VR from ReasonEval) (Reviewer ELDK)
- Reliability of the results:
  - LLaMA 4 Scout 17B Instruct achieve scores comparable to those of top-tier models such as OpenAI o3 (Reviewers ApXz)
  - Employing a simple segmentation (sometimes presumably incorrect) pre-processing that may introduce errors in subsequent evaluation (Reviewer ELDK)
- Lack of comparison between the proposed dataset + evaluation framework and well-known mathematical reasoning benchmarks (Reviewer Ckz3)
- Some novelty concerns (Reviewer ELDK, Ckz3)

**Reviewer Concerns:**

First, the AC considers the use of some outdated LLMs to be a potential concern. While the authors appear to have made a reasonable effort to evaluate models that were recent at the time, mixing older models such as GPT-3.5-turbo (released January 2024) and LLaMA-3 8B (released April 2024) with more recent ones may confound the observed trends. As such, the AC largely agrees with Reviewer ELDK’s concern on this point. That said, evaluation including more recent models is encouraged in future work.

Second, concerns regarding novelty were not the primary focus of all reviews, but were explicitly raised by two reviewers. Reviewer ELDK pointed out in detail that many components of the proposed evaluation framework are not novel, but a combination of them. Reviewer Ckz3 also explicitly raised concerns about limited novelty; while this reviewer acknowledged that the work is primarily a dataset paper, the underlying message about novelty should still be taken. Even setting aside this latter review, the former concern remains valid and, from the AC's perspective, cannot be fully addressed within a single revision cycle.

The further weakness of this work lies in 1) the missing comparison with human experts, and 2) the reliability of the experimental results, both of which were raised by Reviewer ApXz and Reviewer ELDK. First, the authors indeed have made a clear effort to address this issue by incorporating human-expert evaluation results into both the main paper and the appendix. However, this AC believes that the connection to human expert judgments should be more strongly emphasized and integrated into the main narrative. While it is acknowledged that using three human evaluators meets a minimal standard, however, involving more experts would further strengthen the reliability of the conclusions. Regarding the latter issue, the authors do acknowledge this as a limitation in the paper; however, the use of heuristic segmentation may introduce cascading errors that undermine the reliability of the evaluation. This AC recommends that the authors provide a clearer rationale or evidence demonstrating that this preprocessing choice does not materially affect the results. Furthermore, the uniformly low metric values observed in the results raised concerns: even when adopting the authors' perspective, the proposed evaluation does not convincingly differentiate among LLMs and instead yields consistently low scores. The AC agrees with the reviewers that it is problematic and constitutes a significant concern.

Furthermore, the AC agrees with Reviewer Ckz3 that the lack of detailed comparisons with existing mathematical reasoning datasets is an additional concern. Such comparisons would help clarify how CUMath differs from and improves upon prior benchmarks. Finally, after reviewing the paper, the AC finds that the overall presentation (including minor errors and the quality of figures and graphics) would need substantial improvement, as also noted by Reviewer Ckz3.

**Reviewer Scores:**

Two reviewers leaning toward rejection, both at the borderline, unfortunately did not change their stance during discussion; accordingly, this AC does not believe that the paper's score could be increased as a result (please refer to the detailed review for this AC's perspective). Even if the scores have increased, the paper would still require significant refinement. That said, this AC believes that research addressing the challenges of LLM mathematical reasoning remains important and encourages the authors to continue developing this promising line of work.

---

### Decision · Program_Chairs · 2026-01-26

Reject